# Phox2b-expressing neurons contribute to breathing problems in Kcnq2 loss- and gain-of-function encephalopathy models

J. Soto-Perez[1], C. M. Cleary [1], C. R. Sobrinho [1], S. B. Mulkey [2], J. L. Carroll[3], A. V. Tzingounis[1,4] ✉ & D. K. Mulkey [1,4] ✉

Loss- and gain-of-function variants in the gene encoding KCNQ2 channels are a common cause of developmental and epileptic encephalopathy, a condition characterized by seizures, developmental delays, breathing problems, and early mortality. To understand how KCNQ2 dysfunction impacts behavior in a mouse model, we focus on the control of breathing by neurons expressing the transcription factor Phox2b which includes respiratory neurons in the ventral parafacial region. We find Phox2b-expressing ventral parafacial neurons express Kcnq2 in the absence of other Kcnq isoforms, thus clarifying why disruption of Kcnq2 but not other channel isoforms results in breathing problems. We also find that *Kcnq2* deletion or expression of a recurrent gain-of-function variant R201C in Phox2b-expressing neurons increases baseline breathing or decreases the central chemoreflex, respectively, in mice during the light/inactive state. These results uncover mechanisms underlying breathing abnormalities in KCNQ2 encephalopathy and highlight an unappreciated vulnerability of Phox2b-expressing ventral parafacial neurons to KCNQ2 pathogenic variants.

There are five members of the KCNQ (KCNQ1-5; Kv7) family of channels that can function as homo- or hetero-tetramers to produce a sub-threshold K+ conductance that regulates neural firing behavior. Of these, pathogenic variants in KCNQ2 channels are associated with severe developmental and epileptic encephalopathy[1–3]. Most of these variants are loss of function[4,5]; however, more recent work identified a recurrent KCNQ2 pathogenic variant (R201C) that stabilizes the activated state of the channel to produce a gain-of-function effect[6–8]. Patients with the R201C variant display severe neonatal-onset encephalopathy, myoclonus, multifocal seizures later in life, and early profound hypoventilation, developmental problems, and early mortality[9]. Apnea has also been reported in patients with KCNQ2 loss of function but with less severity than gain-of-function variants. Furthermore, KCNQ2 channels are typically co-expressed with KCNQ3

channels[10,11]; despite this, KCNQ3 variants are not associated with respiratory problems. These results show that *KCNQ2* has a complex genotype-phenotype relationship, and despite its physiological significance, the cellular basis for *KCNQ2*-related breathing problems is unknown.

KCNQ2 channels are widely expressed throughout the brain, so disruption of KCNQ2 may disrupt breathing at any level of the respiratory circuit. However, the breathing phenotype associated with the KCNQ2-R201C variant is reminiscent of congenital central hypoventilation syndrome (CCHS)[9], a developmental disorder characterized by severe hypoventilation during sleep and a blunted ventilatory response to $CO_2/H^+$ (i.e., respiratory chemoreception)[12]. Since CCHS is caused by variants in the paired-like homeobox 2b (*Phox2b*) transcription factor that is expressed by several respiratory centers

[1]Dept of Physiology and Neurobiology, University of Connecticut, Storrs, CT, USA. [2]Prenatal Pediatrics Institute, Children's National Hospital, Departments of Neurology and Pediatrics, The George Washington Univ. School of Medicine and Health Sciences, Washington, DC, USA. [3]Dept. of Pediatrics, Univ. Arkansas for Medical Sciences, Little Rock, AR, USA. [4]These authors jointly supervised this work: A. V. Tzingounis, D. K. Mulkey. ✉e-mail: anastasios.tzingounis@uconn.edu; daniel.mulkey@uconn.edu

including chemosensitive RTN neurons[13–15], and since disruption of RTN neurons may contribute to diminished respiratory chemoreception in a mouse model of CCHS[16], we consider RTN neurons a potential substrate responsible for KCNQ2 encephalopathy related breathing abnormalities. This possibility is supported by pharmacological evidence showing that Kcnq channels regulate baseline activity and neuromodulation of RTN neurons in mice[17–19]. However, it remains unclear which Kcnq channel(s) are expressed by RTN neurons and roles of Kcnq2 in the control of breathing have not been determined.

Here, we show that Kcnq2 is the predominant Kcnq isoform in Phox2b-expressing neurons in the ventral parafacial region (i.e., putative RTN neurons). Consistent with previous pharmacological results, we show that conditional ablation of *Kcnq2* from *Phox2b*-expressing neurons (B6(Cg)-Tg(Phox2b-cre)3Jke/J::*Kcnq2*[f/f]; Kcnq2 cKO mice) increases baseline breathing but with minimal effect on the ventilatory response to $CO_2$. Interestingly, we also found that mice expressing *Kcnq2*[R201C/+] in *Phox2b*-expressing neurons (B6(Cg)-Tg(Phox2b-cre)3Jke/J::*Kcnq2*[R201C/+];Kcnq2 GOF mice) breathe normally under baseline conditions but show a blunted ventilatory response to $CO_2$ preferentially during the light/inactive state when inhibition of Kcnq2 channels in RTN neurons by wake-on neurotransmitters is expected to be minimal. At the cellular level, we found that RTN neurons in slices from Kcnq2 GOF mice have a hyperpolarized membrane potential and diminished $CO_2/H^+$ sensitivity, whereas bath application of a Kcnq blocker (ML252) partly rescued the activity deficit of RTN neurons in response to modest stimulations but compromised their ability to respond to more robust depolarizations. These results establish Kcnq2 channels as an important braking mechanism for RTN neural activity. Together these results suggest Phox2b-expressing neurons including those in the ventral parafacial region (putative RTN neurons) are vulnerable to Kcnq2 channelopathies and may contribute to breathing problems associated with KCNQ2 developmental and epileptic encephalopathy (DEE).

## Results

### Phox2b-expressing ventral parafacial neurons preferentially express Kcnq2 transcript

To determine which *Kcnq* isoforms are expressed by putative RTN neurons, we re-analyzed a previously published transcriptomic dataset of ventral parafacial neurons (accession ID GSE174417)[20]. As before[14,15], we identified RTN neurons based on expression of *Slc17a6*, *Phox2b*, *Nmb*, and the absence of *Th*. Cells with this molecular profile have high expression of *Kcnq2* transcript but barely detectable levels of other *Kcnq* subtypes (not shown). In most brain regions, *Kcnq2* typically co-expresses with *Kcnq3* and functions as a heteromeric channel[10,11]. Therefore, to further confirm that *Kcnq2* is the dominant channel subtype expressed by putative RTN neurons, we generated cre-dependent *Phox2b* reporter mice (Phox2b-Cre BAC transgenic mice crossed with a TdTomato [Ai14] reporter mice) and performed fluorescence-activated cell sorting to obtain an enriched population of *Phox2b* expressing neurons for qPCR analysis of all *Kcnq* isoforms. Considering some astrocytes express Phox2b during development[21], we confirmed that our sorted material showed high levels of a neural marker Rbfox3 (NeuN; mean raw CT value $21.97 \pm 0.14$; $n = 3$) and not detectable levels of levels of astrocyte (*Gfap*, *Aldh1l1*) or oligodendrocyte (*Olig1*) specific genes. Consistent with our single-cell RNA sequencing results, we found that *Phox2b*-expressing ventral parafacial neurons have high levels of *Kcnq2* but negligible levels of other *Kcnq* isoforms (Supplementary Fig. 1). We also combine fluorescent in situ hybridization for *Kcnq2* and *Kcnq3* with immunohistochemistry for tdTomato (tdT) fluorescent protein in slices from *Phox2b* reporter mice. We found that 90% of tdT-labeled *Phox2b*-expressing neurons in the ventral parafacial region showed detectable levels of *Kcnq2* transcript, of these 66% lacked *Kcnq3* signal (Fig. 1A–C). Although *Kcnq3* labeling was observed in nearby Phox2b-expressing facial motor

neurons (7N) (Fig. 1A, B), we found minimal colocalization of *Kcnq3* signal with tdT along the ventral surface (Fig. 1A), suggesting *Kcnq3* transcript is sparsely expressed by chemosensitive RTN neurons. We also characterize expression of Kcnq2 and Kcnq3 transcript in Phox2b-expressing neural populations in the locus coeruleus (LC) and nucleus tractus solitarius (NTS). We found abundant expression of both *Kcnq2* and *Kcnq3* mRNA in Phox2b neurons in the LC and NTS (Fig. 1A, B), and in contrast to the RTN, few slices from these regions contained cells that express *Kcnq2* in the absence of *Kcnq3* ($H_3 = 51.59$; $p < 0.0001$) (Fig. 1C). These results suggest *Kcnq2* is the main *Kcnq* subtype in Phox2b-expressing ventral parafacial neurons.

### Kcnq2 deletion or gain-of-function in Phox2b-expressing neurons differentially affect baseline breathing and the central chemoreflex

We first confirmed that in the presence of cre-recombinase mice express one copy of wild-type exon 4 and one copy of mutant exon 4 (Supplementary Fig. 3). To validate the Kcnq2 GOF (Fig. 2A, B) and Kcnq2 cKO models, we prepared brainstem sections containing RTN neurons from Kcnq2[+/+], Kcnq2 GOF and Kcnq2 cKO mice (30–45 days of age) for immunohistochemistry using primary antibodies for Kcnq2 (does not distinguish between Kcnq2 and Kcnq2[R201C] channels) and Phox2b to identify putative RTN neurons. We found the relative proportion of Phox2b-expressing neurons immunoreactive for Kcnq2 was similar between Kcnq2[+/+] and Kcnq2 GOF lines; 59% and 56% of Kcnq2 labeling co-localized with Phox2b in slices from Kcnq2[+/+] and Kcnq2 GOF mice, respectively (Fig. 2C). Approximately a third of Phox2b-expressing ventral parafacial neurons in Kcnq2[+/+] and Kcnq2 GOF tissue lacked Kcnq2 signal and a small proportion of Kcnq2 labeling did not co-localize with Phox2b (Fig. 2C). We also confirmed that Kcnq2-immunoreactivity was largely absent from Phox2b-expressing ventral parafacial neurons in slices from Kcnq2 cKO mice (Fig. 2C). Although the proportion of Kcnq2 GOF (15% of 31 litters) and Kcnq2 cKO (18% of 19 litters) mice in each litter was less than expected (determined within 24 hr of birth), postnatally both genotypes had normal body weight ($p > 0.05$) and life span ($\chi^2 = 4.41$) compared to Kcnq2[+/+] mice ($n = 189$) and floxed only control mice (*Phox2b*[+/+]::*Kcnq2*[R201C/+]; $n = 62$) (Fig. 2D). We also found that Kcnq2[+/+] and Kcnq2 GOF mice showed comparable levels of baseline locomotor activity ($p > 0.05$) and metabolic activity across the light/dark 24-h cycle (Supplementary Fig. 2). Note that insertion of stop-flox R201C may render the allele non-functional in the absence of cre recombinase (see methods); therefore, wild-type and B6(Cg)-Tg(Phox2b-cre)3Jke/J::*Kcnq2*[+/+] mice are used as controls for all experiments. These results suggest loss of *Kcnq2* or expression of one copy of *Kcnq2*[R201C] in *Phox2b*-expressing neurons correlated with early lethality but by weaning age it had minimal impact on survival.

Next, we used whole-body plethysmography to characterize respiratory function in each genotype during the light/inactive and dark/active states. Interestingly, we found that both Kcnq2 cKO and Kcnq2 GOF mice preferentially exhibit breathing abnormalities during the light/inactive state. For example, during the light/inactive state Kcnq2 cKO mice ($n = 9$) show a 49% greater minute ventilation compared to control mice ($n = 15$) under room air conditions ($F_{2,30} = 13.68$, $p < 0.0001$) (Fig. 3D). This phenotype is mediated by an increase in respiratory frequency (Fig. 3B) ($F_{2,30} = 18.51$, $p < 0.0001$) and tidal volume (Fig. 3C) ($F_{2,30} = 4.03$, $p = 0.028$). However, under the same diurnal conditions Kcnq2 cKO mice show an otherwise normal ventilatory response to $CO_2$ (Fig. 3F, G); 0–7% $CO_2$ slope was nearly identical for both genotypes; $p = 0.72$. Unexpectedly, we also found that Kcnq2 cKO mice show normal baseline breathing ($p = 0.97$) (Fig. 4A–D) and central chemoreflex (0–7% $CO_2$ slopes: $0.7 \pm 0.07$ control vs $0.7 \pm 0.04$ Kcnq2 cKO; $p = 0.78$) (Fig. 4E, F) during the dark/active state, thus suggesting the baseline respiratory phenotype of Kcnq2 cKO mice is specific to the light/inactive state. It should be noted that heterozygous deletion of *Kcnq2* in Phox2b-expressing neurons by deletion of

one allele ($Kcnq2^{-/+}$) or by global expression of the $Kcnq2$ knock-in allele ($Kcnq2^{R201C/+}$ mice) in the absence of cre recombinase had negligible effect on baseline breathing ($p = 0.47$ and $p = 0.89$, respectively) or the ventilatory response to $CO_2$ ($p = 0.42$ and $p = 0.97$, respectively). These results are consistent with previous pharmacological data showing that the application of a pan-Kcnq blocker into the RTN of anesthetized rats increased baseline breathing but with minimal effect on $CO_2$/$H^+$-stimulated respiratory output[19].

In contrast to loss of Kcnq2 channel function, we found that Kcnq2 GOF mice breathe normally under baseline conditions but show a blunted ventilatory response to $CO_2$ preferentially during the light/

inactive state. For example, while breathing air in the light/inactive state, control ($n = 8$) and Kcnq2 GOF ($n = 9$) mice show similar respiratory frequency (Fig. 5A, B; $p = 0.14$), tidal volume (Fig. 5A, C; $p = 0.29$) and minute ventilation (Fig. 5A–D; $p = 0.94$). KCNQ2 GOF mice also exhibit sighs (augmented breaths which help maintain lung compliance) at a frequency similar to control mice during both light and dark cycles ($F_{3,18} = 0.53$; $p = 0.66$). We also used Poincare analysis to assess apnea propensity on a breath-by-breath basis and respiratory stability over an expanded time scale. Since the RTN is an important determinant of breathing during sleep when apneas more frequently occur[22], we focused this analysis on respiratory frequency (-10,000

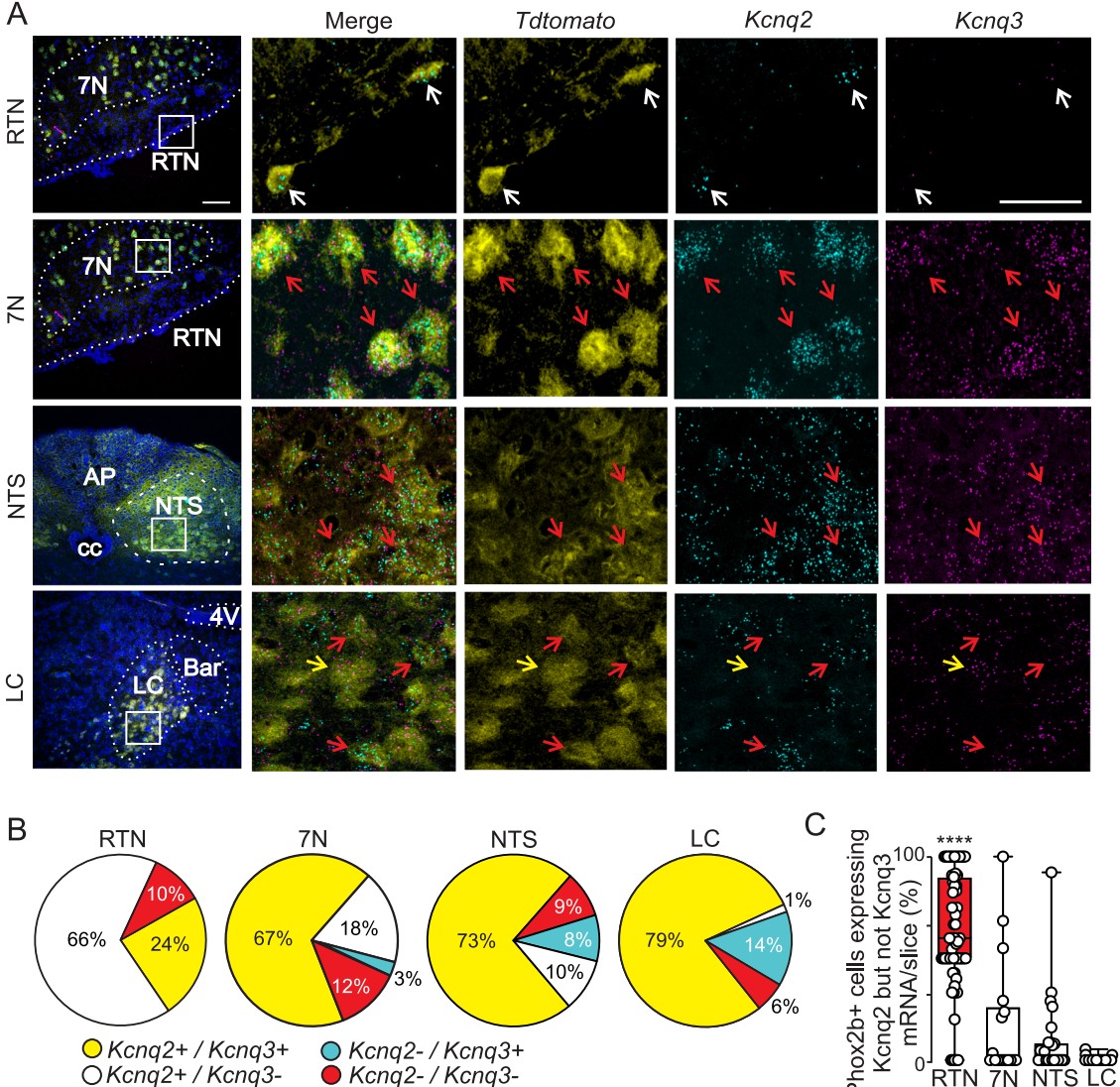

**Fig. 1 | *Phox2b*-expressing RTN neurons preferentially express *Kcnq2* but not *Kcnq3* channels.** Coronal sections from $Phox2^{Cre/+}$::Ai14 reporter mice were obtained for multiplex in situ hybridization analysis of Kcnq2 and Kcnq3 transcript expression in for Phox2b-expressing brainstem populations. **A** Photomicrographs of coronal sections from the RTN, 7N, NTS, and LC show tdT-labeled *Phox2b+* neurons (yellow) in the RTN express *Kcnq2* transcript (cyan) but low *Kcnq3* signal (magenta). Insets (right), shows fluorescent in situ hybridization results for the boxed region at a higher magnification where *Kcnq2, Kcnq3*, and *Phox2b*-tdT (yellow), the DAPI (blue). Signal was filtered to improve visualization. White arrows identify cells that express *Kcnq2* but not *Kcnq3*, red arrows identify cells that co-express both transcripts, and yellow arrows designate cells that express *Kcnq3* but not *Kcnq2*. Scale bars are 100 μm and 50 μm (insets). **B** Summary of fluorescent in-situ hybridization results ($n = 3$ mice, 30–40 days postnatal) show that 90% of tdT-*Phox2b* labeling in the ventral parafacial region co-localized with *Kcnq2* labeling

(white) and of these only 24% also showed *Kcnq3* labeling (yellow), whereas 10% of tdT-*Phox2b* neurons in this region lacked both *Kcnq2* and *Kcnq3* signal (red). In contrast to the RTN, the majority of *Kcnq2* signal detected in the *7N, LC and NTS* co-localized with *Kcnq3* labeling and in some cases only *Kcnq3* labeling was observed (cyan). **C** Summary data plotted as mean ± maximum and minimum show the proportion of Phox2b-expressing cells per slice that express *Kcnq2* transcript in the absence of *Kcnq3* in the RTN ($n = 66$ slices, red), 7N ($n = 17$ slices, white), NTS ($n = 23$ slices, white), LC ($n = 16$ slices, white). Significance was determined using the Kruskal-Wallis test; RTN vs NTS ($p < 0.0001$), RTN vs LC ($p < 0.0001$), RTN vs. 7N ($p < 0.0001$). These results show that Phox2b-expressing neurons in the ventral parafacial region have a higher proportion of *Phox2b+* neurons that uniquely express *Kcnq2* mRNA than other medullary *Phox2b*-expressing neural populations. 7N, facial motor nucleus; LC, locus coeruleus; nucleus tractus solitarius (NTS).

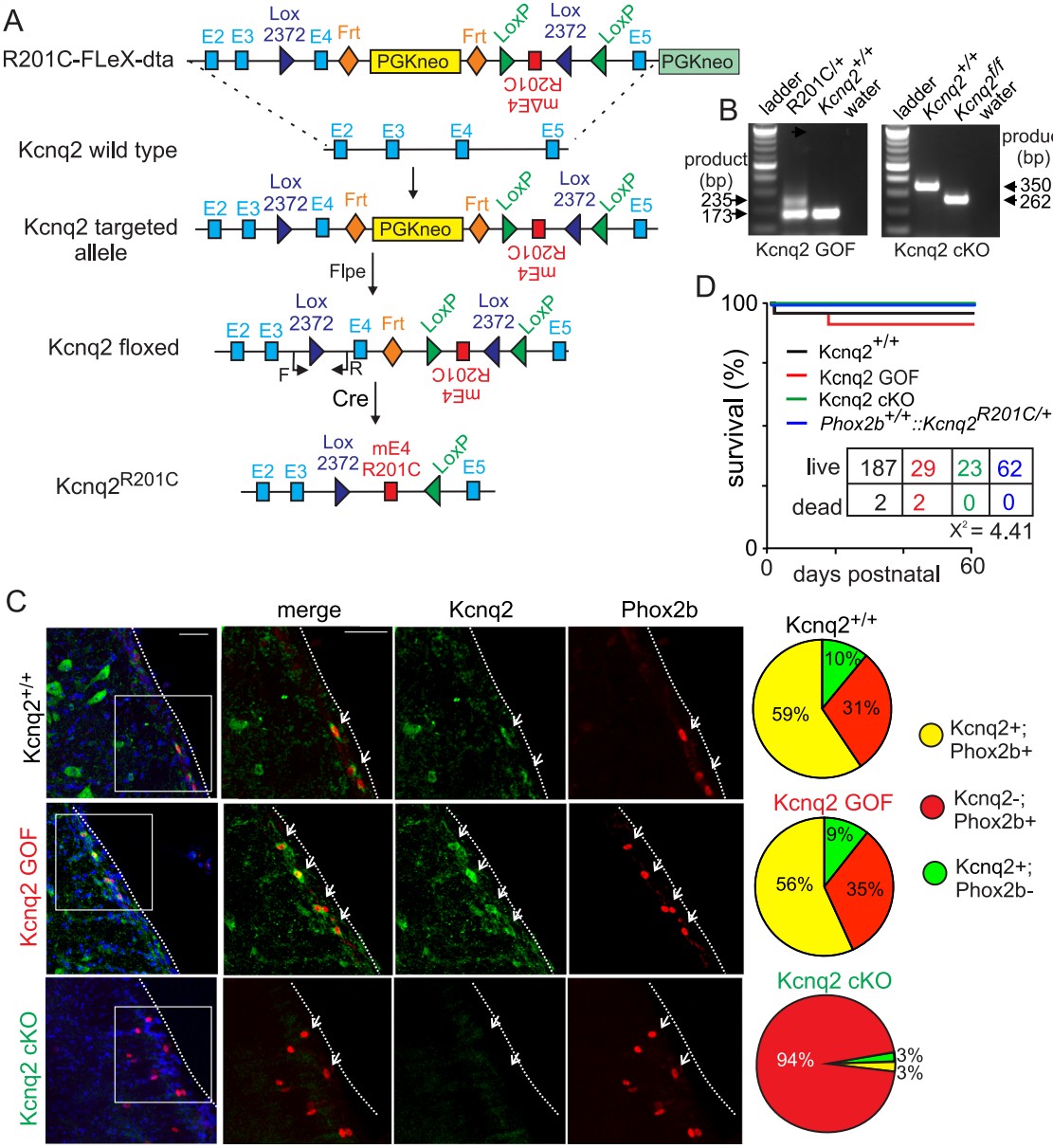

**Fig. 2 | Characterization of Kcnq2 cKO and Kcnq2 GOF mouse models.**
**A** Schematic of *Kcnq2* wild-type locus and targeting construct containing the inverted mutated exon 4 (mE4). When Cre recombinase is expressed, wild type exon 4 is removed and mE4 exon is inverted to effectively replace wild-type exon 4 with one expressing R201C. We crossed the Kcnq2 floxed line with mice that express Cre under control of the *Phox2b* promotor to express *Kcnq2^R201C* conditionally (B6(Cg)-Tg(Phox2b-cre)3Jke/J::*Kcnq2^R201C/+*; Kcnq2 GOF). **B** Genotyping PCR analysis for Kcnq2 GOF and Kcnq2 cKO mice. The PCR products run to the expected sizes for each genotype and primer set (Kcnq2 GOF primers span exon 4, including residue 201 of exon 4; Kcnq2 cKO primers include a loxP site). Water was used as a no-template negative control. **C**, Immunohistochemistry was performed to characterize expression of Kcnq2 protein in Phox2b-expressing ventral parafacial neurons in brainstem sections from Kcnq2^+/+, Kcnq2 GOF, and Kcnq2 cKO mice. Images of coronal medullary sections (-6.24 mm behind bregma) from Kcnq2^+/+ and Kcnq2 GOF mice show robust Kcnq2 signal (green) co-localized with Phox2b

labeling (red); and confirm reduction of Kcnq2 protein in Phox2b-expressing ventral parafacial neurons in sections from Kcnq2 cKO mice. Right, summary of immunohistochemistry results shows the relative proportions of Phox2b-expressing neurons in the ventral parafacial region is comparable between Kcnq2^+/+ ($N = 3$ animals; $n = 1240$ cells), Kcnq2 GOF ($N = 3$ animals, $n = 956$ cells), and Kcnq2 cKO ($N = 2$ animals, $n = 486$ cells) mice. We also found similar proportions of Phox2b-expressing cells in sections from Kcnq2^+/+ (59%) and Kcnq2 GOF (57%) mice are Kcnq2-immunoreactive (Kcnq2-IR), whereas approximately a quarter of Phox2b-expressing cells did not show detectable levels of Kcnq2, and a relatively modest level of Kcnq2-IR was also observed in Phox2b-negative cells. As expected, virtually all (96%) Phox2b-expressing neurons in the ventral parafacial region in sections from Kcnq2 cKO mice lack expression of Kcnq2. Scale bar 50 μm. **D** survival curves for each experimental group as well as floxed-only control mice (*Phox2b^+/+::Kcnq2^R201C/+*) show that each genotype exhibits normal survival. These results were compared using Kaplan-Meier survival curve comparison.

breaths) measured during 20 min periods in air during both light/inactive and dark/active states. Under these conditions, we found that control littermates and Kcnq2 GOF mice showed similar frequency (spontaneous $F_{3,18} = 0.34$, $p = 0.79$; post-sigh $F_{3,18} = 1.49$, $p = 0.25$) and duration (spontaneous $F_{3,17} = 0.16$, $p = 0.92$; post-sigh $F_{3,16} = 0.52$, $p = 0.67$) of apneic events. These results show that expression of

Kcnq2^R201C in Phox2b-expressing neurons did not perturb baseline breathing. However, consistent with the voltage-dependent nature of Kcnq2 channels and their contribution to stimulated neural activity, we found that Kcnq2 GOF mice have a diminished capacity to increase minute ventilation in response to graded increases in $CO_2$ from 0% to 7% (Fig. 3G); 0–7% $CO_2$ slope: $0.7 \pm 0.06$ control vs $0.5 \pm 0.04$ Kcnq2

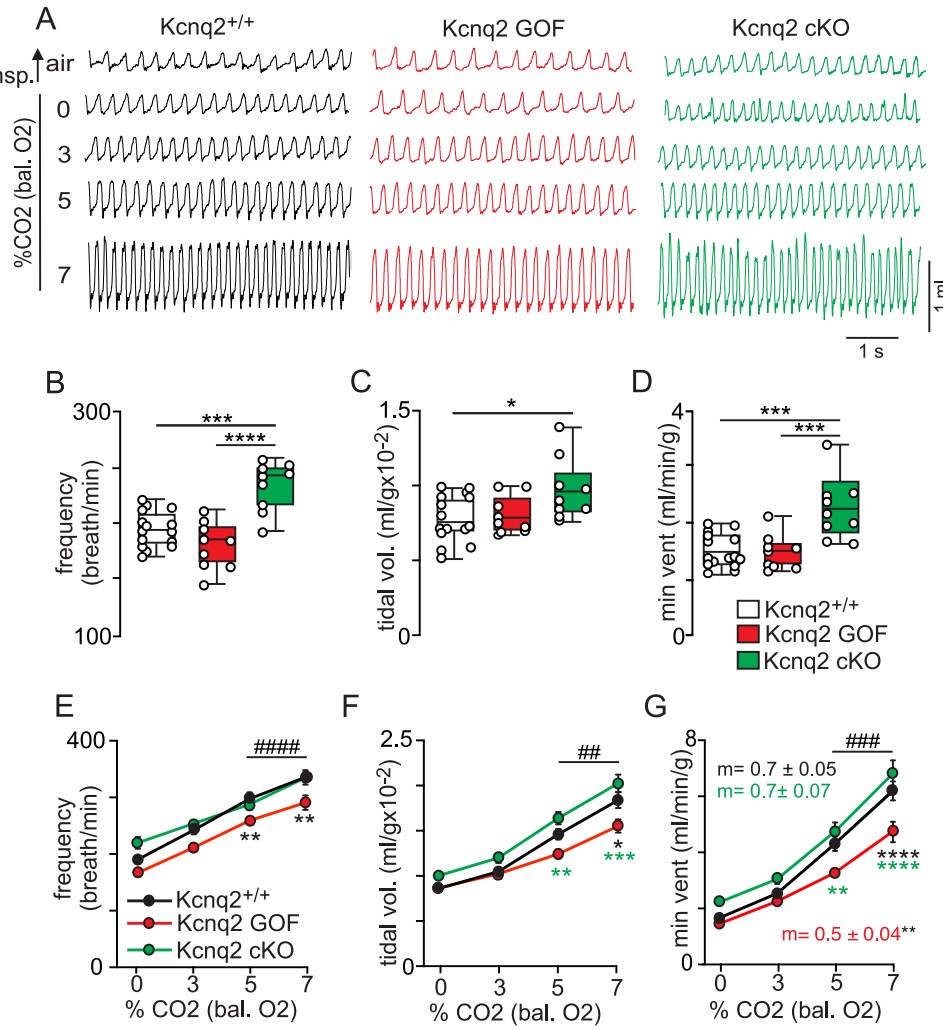

**Fig. 3 | Loss and Gain of Kcnq2 function in Phox2b neurons differentially affects baseline breathing and central chemoreception. A** Traces of respiratory activity from Kcnq2$^{+/+}$ (white), Kcnq2 GOF (red), and Kcnq2 cKO (green) mice during exposure to room air and graded increases in $CO_2$ (0–7%; balance $O_2$). Summary data ($n = 15$ Kcnq2$^{+/+}$; $n = 9$ Kcnq2 GOF; $n = 9$ Kcnq2 cKO) plotted as mean ± maximum and minimum show respiratory frequency (**B**; $p < 0.0001$), tidal volume (**C**; $p = 0.0282$) and minute ventilation (**D**; $p < 0.0001$) are increased under room air conditions in Kcnq2 cKO mice compared to control or Kcnq2 GOF mice (one-way ANOVA and Tukey's multiple comparison test). Summary data (Kcnq2$^{+/+}$ $n = 15$; Kcnq2 GOF $n = 9$; Kcnq2 cKO $n = 9$) plotted as mean ± SEM of frequency (**E**; $p < 0.0001$), tidal volume (**F**; $p < 0.0001$) and minute ventilation (**G**; $p < 0.0001$) show that Kcnq2 GOF mice have a reduced capacity to increase respiratory output in response to graded increases in $CO_2$. Means are compared using two-way ANOVA followed by Tukey's multiple comparison test and slope of the minute ventilation 0–7% $CO_2$ response is compared using one-way ANCOVA. Asterisk (*) indicates the difference between genotypes; # used to distinguish within genotype differences from control. One symbol = $p < 0.05$, two symbols = $p < 0.01$, three symbols = $p < 0.001$, four symbols = $p < 0.0001$.

GOF; p = 0.0013. This central chemoreflex deficit was only evident in the light/inactive state (Fig. 5E) but not the dark/active state (Fig. 5F); 0–7% $CO_2$ slope: 0.9 ± 0.07 control vs 0.8 ± 0.06 Kcnq2 GOF; $p = 0.26$. These results are consistent with the possibility that output of RTN neurons is a main determinant of breathing during sleep[22]. It is also interesting to note that Kcnq channels in RTN neurons are inhibited by several wake-on transmitters including serotonin[18,19], acetylcholine[17], and norepinephrine[23]; therefore, we speculate that inhibition of Kcnq2 by these transmitters limits the impact of Kcnq2 mutations (deletion or GOF) on respiratory output during wakefulness. Importantly, the respiratory phenotype exhibited by Kcnq2 GOF mice appears specific to a central chemoreception deficit since adult Kcnq2 GOF mice show a normal ventilatory response to hypoxia (10% $O_2$; balance $N_2$) (hypoxia increased activity by 1.1 ± 0.3 and 1.4 ± 0.3 ml/g/min in control and Kcnq2 GOF, respectively; unpaired t test, $p = 0.43$). Together, these results suggest that disruption of Kcnq2 channel function in Phox2b-expressing neurons disrupts central chemoreception and may contribute to breathing problems associated with Kcnq2 encephalopathy.

## Effects of KCNQ2$^{R201C/+}$ on activity of RTN neurons

Phox2b is expressed by several respiratory centers[24] including RTN neurons that function as important respiratory chemoreceptors[13]; therefore, we wanted to determine whether and how expression of Kcnq2$^{R201C/+}$ affects activity of RTN neurons. To make this determination, we made cell-attached current-clamp recordings to characterize activity of RTN neurons in slices from neonatal control and Kcnq2 GOF mice under control conditions and during exposure to 10% $CO_2$. RTN neurons were identified based on location within the ventral parafacial region (Fig. 6A), a minimum 0.5 Hz excitatory response to 10% $CO_2$ (note that for non-spontaneous cells a positive DC current was delivered to elicit a background level of activity and this current was maintained for the duration of a $CO_2$ test), and in some cases Phox2b immunoreactivity ($n = 11$ neurons from control mice, $n = 10$ neurons from Kcnq2 GOF mice) or *Phox2b*-dependent tdT expression ($n = 21$ neurons from Kcnq2 GOF mice) (Fig. 6A). RTN neurons in slices from control mice identified in this manner show a $CO_2/H^+$ response profile similar to what we[20] and others[25] have reported for RTN neurons;

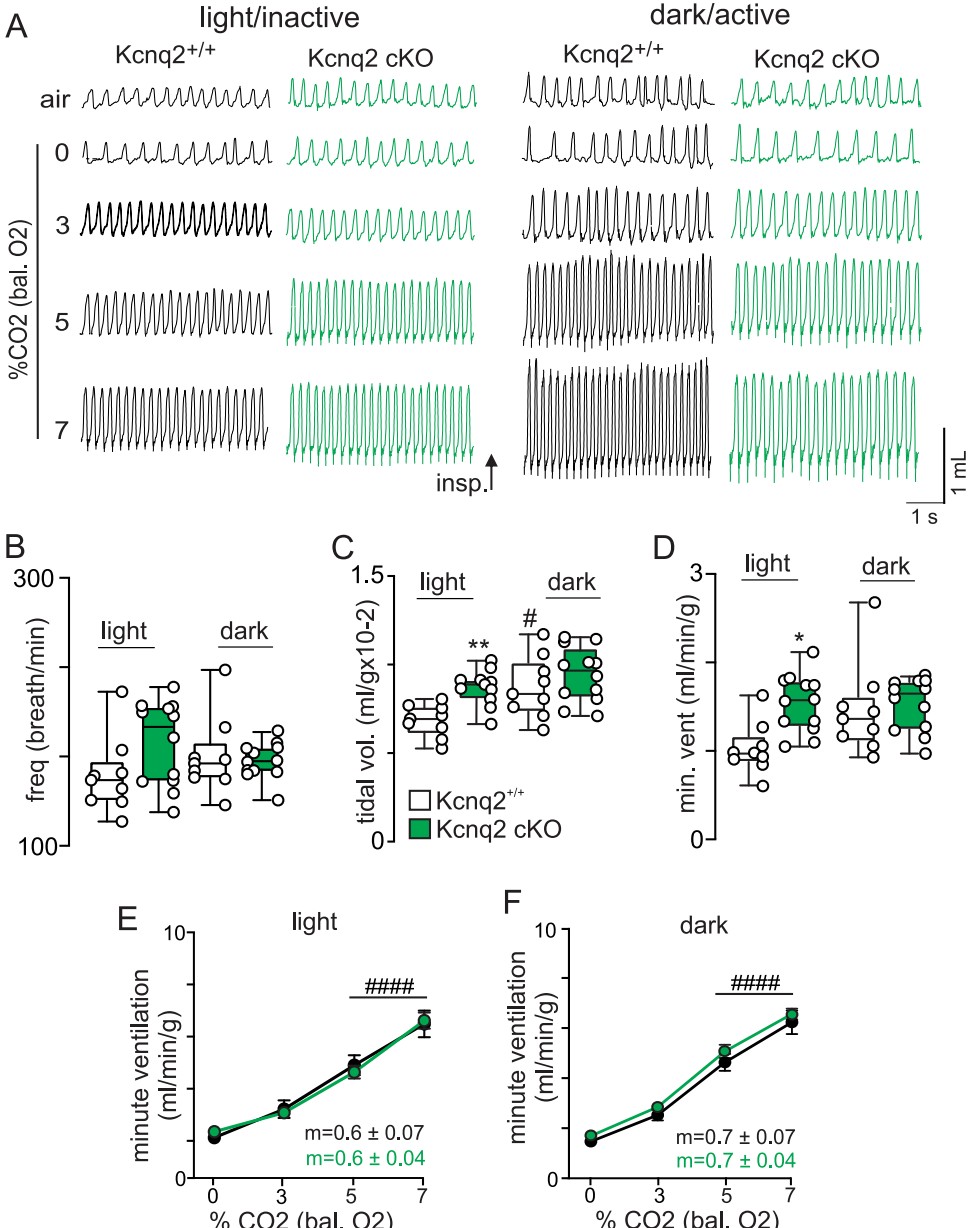

**Fig. 4 | Kcnq2 cKO mice show increased baseline breathing preferentially during the light/inactive state. A** Traces of respiratory activity from Kcnq2[+/+] (white) and Kcnq2 cKO (green) mice in room air, 100% $O_2$ and 3–7% $CO_2$ (balance $O_2$) during the light/inactive and dark/active states. Summary data ($n = 9$ Kcnq2[+/+] and $n = 12$ Kcnq2 cKO mice) plotted as mean ± maximum and minimum frequency (**B**; $p = 0.2180$), tidal volume (**C**; $p = 0.0009$) and minute ventilation (**D**; $p = 0.0187$) shows that Kcnq2 cKO mice exhibit higher baseline respiratory activity during the light/inactive state but not during dark/active conditions. Summary data (Kcnq2[+/+]

$n = 9$ and Kcnq2 cKO $n = 12$ mice) plotted as mean ± SEM of minute ventilation show that Kcnq2[+/+] and Kcnq2 cKO mice exhibit similar ventilatory responses to $CO_2$ during the light/inactive (**E**, $p = 0.9621$) and dark/active states (**F**, $p = 0.0717$). Means are compared using two-way ANOVA followed by Tukey's multiple comparison test and the slope of the minute ventilation 0–7% $CO_2$ response are compared using one-way ANCOVA. Asterisk (*) indicates the difference between genotypes and # designates within genotype differences from control; one symbol = $p < 0.05$, two symbols = $p < 0.01$, three symbols = $p < 0.001$, four symbols = $p < 0.0001$.

spontaneous activity under control conditions of 5% $CO_2$ ($1.8 \pm 0.25$ Hz; $N = 24$) and activated 10% $CO_2$ ($CO_2$ increased activity by $1.7 \pm 0.2$ Hz, $N = 24$) (Fig. 6C–E). However, approximately half of RTN neurons in slices from Kcnq2 GOF mice were not spontaneously active under control conditions and those that where spontaneously active were less active under control conditions ($0.8 \pm 0.1$ Hz, $F_{2,49} = 8.97$, $p = 0.0005$) and showed a blunted response to 10% $CO_2$ compared to RTN neurons in control tissue ($CO_2$ increased activity by $1.0 \pm 0.1$ Hz, $F_{2,50} = 10.34$, $p = 0.0002$) (Fig. 6B–D). Furthermore, to bolster the possibility that wake-on transmitter signaling limits the impact of Kcnq2 GOF on output of RTN neurons, we also characterized serotonin sensitivity of RTN neurons in slices from each genotype. We found that

bath application of 5 μM serotonin (5HT) increased activity of RTN neurons in slice from Kcnq2[+/+] ($n = 7$) and Kcnq2 GOF ($n = 5$) by $2.0 \pm 0.3$ ($T_6 = 6.06$, $p = 0.0009$) and $1.8 \pm 0.3$ Hz ($T_4 = 6.3$, $p = 0.003$), respectively, and negated differences in activity between genotypes ($p = 0.51$) (Fig. 6E, F). To characterize $CO_2$/H[+] sensitivity of non-spontaneous cells, we delivered a depolarizing current to elicit a baseline activity level of ~1 Hz. Under these conditions, a subset of non-spontaneous cells responded to 10% $CO_2$ by an amount similar to spontaneous RTN neurons in Kcnq2 GOF tissue ($CO_2$ increased activity by $0.9 \pm 0.1$ Hz) (Fig. 6C, E). Neurons that did not show a minimum $CO_2$ response (even after adjusting baseline activity) were excluded from further analysis. These results show that expression of Kcnq2[R201C] in

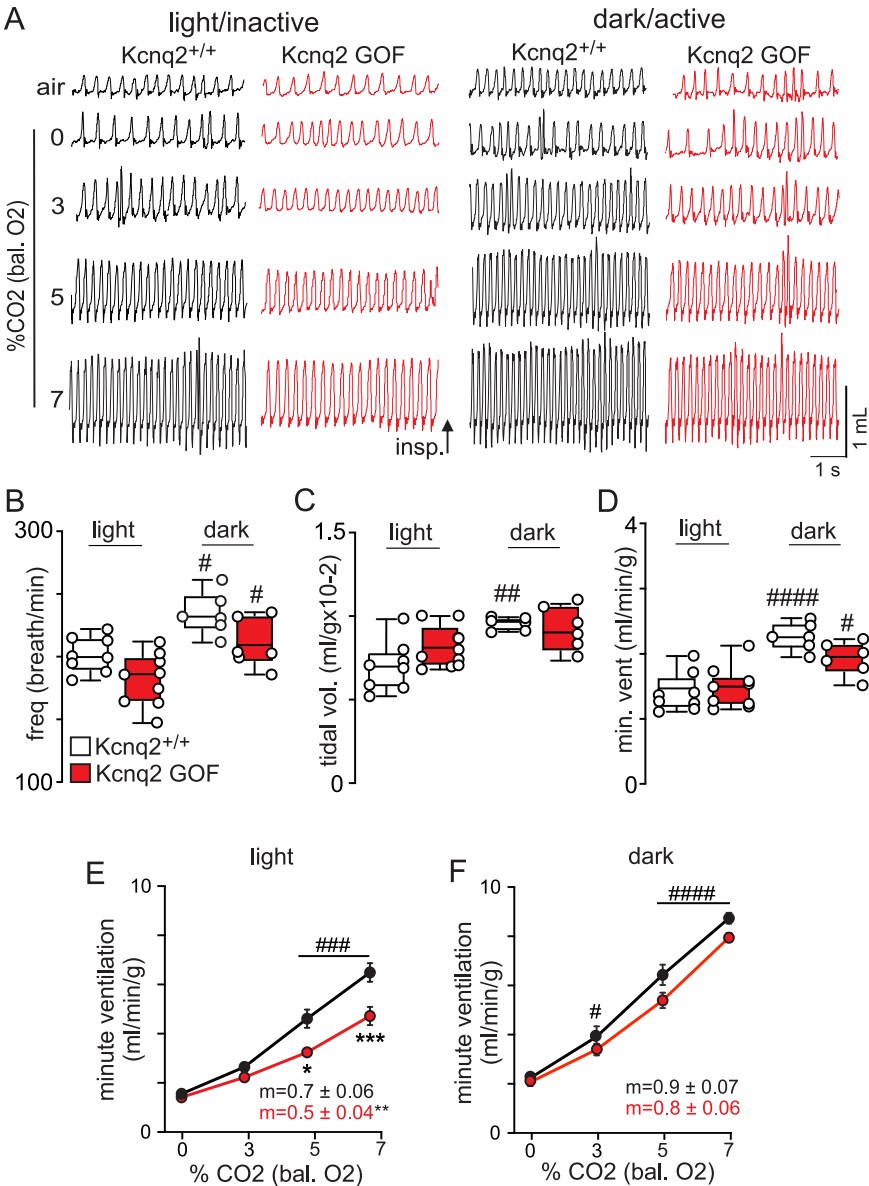

**Fig. 5 | Kcnq2 GOF mice show a blunted ventilatory response to CO₂ during the light/inactive state. A** Traces of respiratory activity from Kcnq2[+/+] (white) and Kcnq2 GOF (red) mice in room air, 100% $O_2$ and 3–7% $CO_2$ (balance $O_2$) during the light/inactive and dark/active states. Summary data (light/inactive n = 8 mice/genotype; dark/active n = 6 mice/genotype) plotted as mean ± maximum and minimum respiratory frequency (**B**; p < 0.0001), tidal volume (**C**; p = 0.0021) and minute ventilation (**D**; p < 0.0001) show that both genotypes exhibit a characteristic arousal-dependent increase in respiratory activity in the dark/active state and no genotype differences were observed under room air conditions. Summary data (light/inactive n = 8 mice/genotype; dark/active n = 6 mice/genotype) plotted as mean ± SEM of minute ventilation show that $CO_2/H^+$-dependent respiratory output in Kcnq2 GOF mice is suppressed in the light/inactive state (**E**, same data as shown in Fig. 3g, p < 0.0001) but not during the dark/active state where the slope of the ventilatory response to $CO_2/H^+$ was similar to control (p = 0.27). Means are compared using two-way ANOVA followed by Tukey's multiple comparison test and slope of the minute ventilation 0–7% $CO_2$ response are compared using one-way ANCOVA. Asterisk (*) indicates the difference between genotypes and # designates within genotype differences from control; one symbol = p < 0.05, two symbols = p < 0.01, three symbols = p < 0.001, four symbols = p < 0.0001.

Phox2b-expressing neurons suppressed baseline firing and $CO_2/H^+$-stimulated activity of RTN neurons.

This activity deficit was also evident in whole-cell current-clamp mode where $CO_2/H^+$ activated RTN neurons in slices from Kcnq2 GOF mice fired fewer action potentials in response to depolarizing current steps from a holding potential of −65 mV ($T_{17}$ = 3.11, p = 0.006) (Fig. 7A, B). Consistent with our hypothesis, bath application of a pan-Kcnq channel blocker (ML252; 10 μM) improves the firing response of RTN neurons in Kcnq2 GOF tissue to modest depolarizing current steps (25–50 pA) to the point of being similar to control (p = 0.82). However, during more pronounced current injections (≥75 pA) RTN neurons from both genotypes showed pronounced spike amplitude and frequency decrement (depolarizing block) in the presence of ML252 (Fig. 7A, B). This result further demonstrates the requisite role of Kcnq2 channels in limiting RTN stimulated activity. We also found that RTN neurons in slices from Kcnq2 GOF mice had a hyperpolarized resting membrane ($F_{3,27}$ = 10.48; p < 0.0001) (Fig. 7C) but similar input resistance ($F_{3,25}$ = 2.38; p = 0.09) (Fig. 7D) compared to control neurons. These results are consistent with the well-established voltage-dependent properties of Kcnq2 channels. These results also identify Kcnq2 channels as a potential therapeutic target for breathing problems in *Kcnq2* encephalopathy. To test this possibility, we treated control and Kcnq2 GOF mice (N = 6/genotype) with a single dose of ML252 (30 mg/kg; dissolved in DMSO and diluted in saline for I.P.

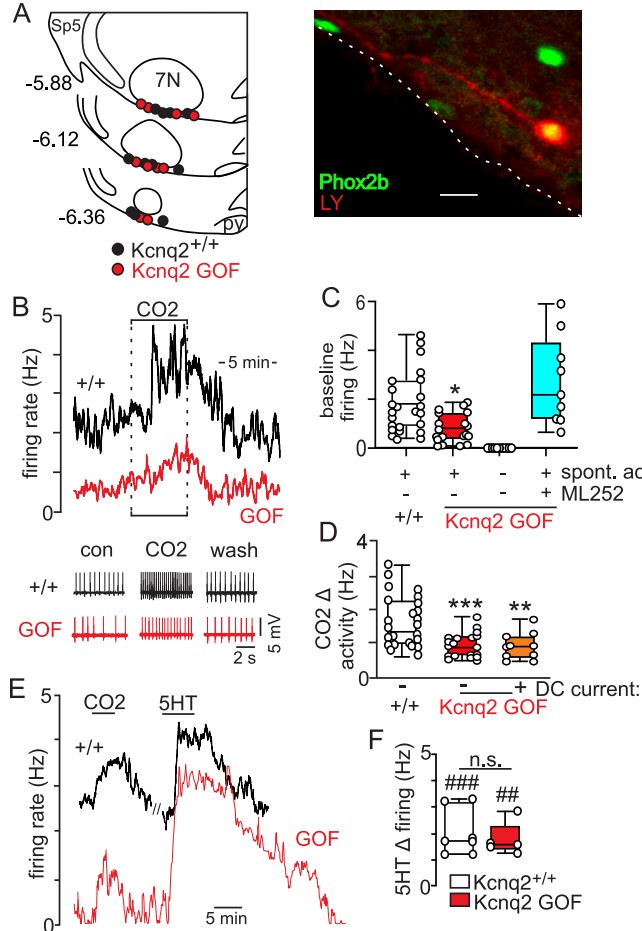

**Fig. 6 | RTN neurons in slices from Kcnq2 GOF mice show reduced baseline activity and $CO_2/H^+$ sensitivity. A** Computer-assisted plot shows the location of RTN neurons from each genotype in the ventral parafacial region. Sp5, spinal trigeminal nucleus. Right, double-immunolabeling shows a Lucifer Yellow (LY)-filled $CO_2/H^+$ sensitive RTN neuron recorded in a Kcnq2 GOF slice is immunoreactive for Phox2b (green). We confirmed Phox2b immunoreactivity in RTN neurons from control (n = 11) and Kcnq2 GOF ($n = 10$) tissue. Numbers to the left of each section designate millimeters from bregma. Scale bar: 20 μm. 7N, facial motor nucleus. **B** Traces of firing rate and segments of membrane potential (spikes are truncated) from chemosensitive RTN neurons in slices from Kcnq2$^{+/+}$ (black) and Kcnq2 GOF (red) mice show examples of spontaneous activity under control conditions (5% $CO_2$; pH 7.3) and that neurons from both genotypes respond to 10% $CO_2$ (pH 7.0). Summary data shows that all RTN neurons in control tissue (n = 24 cells; white) are spontaneously activated (**C**) and respond to 10% $CO_2$ with a robust increase in firing (**D**). Conversely, only approximately half of RTN neurons in slices from Kcnq2 GOF mice are spontaneously active (n = 19, 66%; red) under control conditions (**C**) and both spontaneous and non-spontaneous (with baseline adjusted to -1 Hz by application of a DC current; orange) RTN neurons from Kcnq2 GOF slices show a diminished firing response to $CO_2$ compared to RTN neurons from control tissue (**D**; $p = 0.0002$) (One-Way ANOVA with Tukey's multiple comparison test). **E, F**, Firing rate traces and summary data (plotted as mean ± maximum and minimum) show responses of chemosensitive RTN neurons in slices from Kcnq2$^{+/+}$ (black; n = 7 cells) and Kcnq2 GOF (red; n = 5 cells) mice to 10% $CO_2$ and serotonin (5 HT; 10 μM). We found that exposure to 5HT increased activity of RTN neurons in slices from Kcnq2$^{+/+}$ and Kcnq2 GOF to an amount that was not different between genotypes ($p = 0.5837$; one-sided, non-parametric Mann-Whitney test).

injection) followed 30 min later by assessment of respiratory function. Contrary to expectations, we found that systemic injection of ML252 had no discernible effect on breathing in control or Kcnq2 GOF mice (minute ventilation $p = 0.062$). However, in the absence of a positive control, we interpret these results with caution and remain optimistic

that Kcnq2 channels can be targeted by systemic drug application to improve respiratory output.

## Discussion

*Kcnq2* is one of the most commonly mutated genes in developmental and epileptic encephalopathy. The emerging *Kcnq2* genotype-phenotype relationship suggests loss of function variants result in neonatal epilepsy, whereas gain-of-function variants are associated with more severe symptoms including profound developmental delay, severe hypoventilation and early mortality[4]. To better understand this relationship at the cellular level, we manipulated Kcnq2 function in Phox2b-expressing neurons and characterized respiratory function at the cellular and whole animal levels. We show that both loss and gain of Kcnq2 function in Phox2b-expressing neurons preferentially disrupted baseline breathing or the central chemoreflex, respectively, during the light/inactive state. Also, expression of Kcnq2$^{R201C}$ in Phox2b neurons suppressed baseline activity and $CO_2/H^+$ sensitivity of RTN neurons. These results suggest Kcnq2 channels regulate breathing in a diurnal-dependent manner and identify Phox2b-expressing neurons including RTN neurons as potential candidates for breathing problems in *Kcnq2* encephalopathy.

Kcnq2 channels are a voltage-dependent delayed-rectifying K$^+$ channel that activates at subthreshold conductance to influence resting membrane potential and basal firing behavior[26,27]. Also, since Kcnq2 conductance increases with depolarization and since these channels do not readily inactivate, they effectively limit stimulated activity in response to high $CO_2$ or wake-on transmitters. Kcnq2 channels can function as homotetramers; however, when co-expressed with other Kcnq2 channels including Kcnq3 or Kcnq5 they preferentially function as heterotetramers[28,29]. Previous work also showed that Kcnq2 is co-expressed with Kcnq3 or Kcnq5 in other respiratory centers including inspiratory rhythmogenic pre-Bötzinger complex neurons[30,31] where it contributes to inspiratory burst termination[32] and hypoglossal motor neurons[33] to control respiratory motor output. Therefore, it is reasonable to expect variants in any of these channel subtypes to have a similar negative impact on control of breathing. However, this is not the case. For example, although multiple groups have identified orthologous Kcnq3 gain of function variants in patients with autism spectrum disorders or electrical status epilepticus in sleep[7,34], these patients do not show evidence of disordered breathing. Similarly, Kcnq5 gain-of-function variants are associated with intellectual disability with or without epilepsy but no breathing phenotypes have been reported[35–37]. It is not clear why disruption to Kcnq3 or Kcnq5 is not associated with breathing problems. From an RTN-centric perspective, the lack of Kcnq3 (Fig. 1) and Kcnq5 (Supplementary Fig. 4) transcript in Phox2b-expressing ventral parafacial neurons is one possibility. Also, considering Kcnq3 is typically expressed with Kcnq2 in other respiratory areas including the pre-bötzinger complex[30,31], hypoglossal motor nucleus[33], LC and NTS (Fig. 1), and since mutant and wild-type channels can form heteromeric channels[4,6], it is reasonable to predict the impact of mutant Kcnq3 or possibly Kcnq5 will be reduced by heteromerization with wild type Kcnq2. By similar logic, our results show that Phox2b-expressing ventral parafacial neurons preferentially express Kcnq2 in the absence of other channel binding partners (Fig. 1, supplementary Figs. 1e and 4); therefore, we suspect this renders RTN neurons and respiratory function susceptible to Kcnq2 dysfunction, and consequently, a likely basis for breathing problems in Kcnq2 encephalopathy.

Based on our evidence that homomeric Kcnq2 channels are preferentially expressed by Phox2b-expressing ventral parafacial neurons, it is unclear why *Kcnq2* haploinsufficiency has unremarkable effects on breathing. This finding is consistent with clinical evidence showing patients with *KCNQ2* haploinsufficiency are not reported to exhibit disordered breathing[38]. One explanation for this is that neuronal depolarization also activates additional K$^+$ currents, most notably

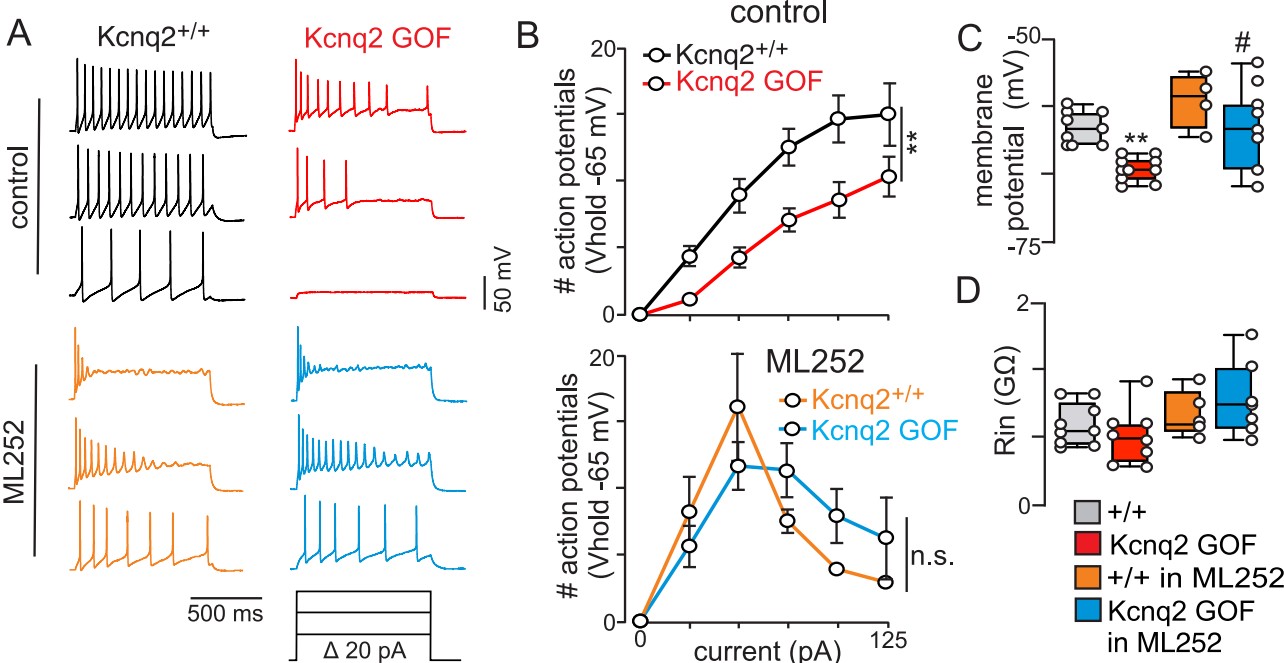

**Fig. 7 | Expression of Kcnq2 GOF limits repetitive firing of RTN neurons whereas pharmacological inhibition of Kcnq2 favors depolarizing block. A** Segments of membrane potential from RTN neurons in slices from Kcnq2[+/+] (black) and Kcnq2 GOF (red) mice during depolarizing current injections (0 to +125pA, 1 s duration) from a membrane potential of −65 mV under control conditions and in the presence of a selective Kcnq2 channel blocker (ML252; 10 μM; Kcnq2[+/+] + ML252 = orange; Kcnq2 GOF + ML252 = Cyan). **B** Input-output relationships (plotted as mean ± SEM) show that under control conditions (top) RTN neurons in slices from Kcnq2 GOF mice (n = 11) generate fewer action potentials in response to depolarizing current injection compared to RTN neurons from control tissue (n = 8) (p = 0.0063; area under curve was compared by unpaired T-test). Note also that bath application of ML252 (10 μM) normalized the response of RTN neurons from each genotype to modest depolarizing current injections (up to 50 pA) but resulted in depolarizing block at more positive steps (**B**, bottom, p = 0.8162). **C** Summary data (plotted as mean ± maximum and minimum) show that RTN neurons from Kcnq2 GOF mice (n = 10) have a hyperpolarized resting membrane potential compared to RTN neurons from Kcnq2[+/+] mice (n = 9) under control conditions but not in the presence of ML252 (10 μM) (F$_{3,27}$ = 10.48; p < 0.0001; one-way ANOVA with Tukey's multiple comparison test). **D** Summary results (plotted as mean ± maximum and minimum) show that input resistance measured during −100 pA step was similar between genotypes (Kcnq2[+/+] n = 9; Kcnq2 GOF n = 8) under control conditions and in the presence of ML252 (10 μM; Kcnq2[+/+] + ML252 n = 5; Kcnq2 GOF + ML252 n = 7; one-way ANOVA with Tukey's multiple comparison test).

Ca[2+]-activate K[+] currents. Indeed, we have shown that blocking Kcnq channels leads to the activation of an apamin-sensitive Ca[2+]-activate K[+] conductance that also serves to limit the stimulated activity of RTN neurons[19]. Hence, loss of Kcnq channel activity may be compensated by activation of these or other voltage-gated K[+] channels resident in RTN neurons. We expect contributions of voltage-dependent K[+] channels to be more evident during CO$_2$/H[+] stimulated activity. Consistent with this, previous work from anesthetized rats that showed RTN injections of a pan-Kcnq channel blocker (XE991) increased baseline breathing and lowered the CO$_2$ apneic threshold but did not potentiate the ventilatory response to a maximum stimulus of 10% CO$_2$[19]. Further, we show here that Kcnq2 cKO mice show enhanced respiratory output under baseline (air) conditions but an otherwise normal respiratory response to CO$_2$ (Figs. 3–4). On the other hand, increasing Kcnq channel activity either through the expression of a GOF variant such as Kcnq2[R201C] or pharmacologically with the pan-Kcnq activator retigabine can hyperpolarize membrane potential below firing threshold (Fig. 7C) and limit activity of RTN neurons (Figs. 6–7).

Our evidence that Kcnq2 loss or gain of function in Phox2b-expressing neurons disrupted breathing during the light/inactive but not dark/active state is consistent with previous work showing that injection of XE991 (pan Kcnq channel blocker) into the RTN of anesthetized rats increased baseline breathing, where this same manipulation in conscious rats had negligible effect on breathing[19]. The basis for potential circadian differences in behavioral responses to Phox2b neuron-specific manipulations of Kcnq2 is not clear. One possible explanation is that RTN neurons have a dominant role in control of breathing during times of reduced arousal, whereas during

wakefulness contributions of RTN neurons to breathing may be diluted by numerous other excitatory inputs to the respiratory system. Consistent with this, targeted ablation of RTN neurons using a newly developed Nmb[Cre/+] mouse resulted in hypoventilation under all vigilance states, but most profoundly during NREM sleep[39]. Another possibility is that inhibition of Kcnq2 channels by wake-on neuromodulators limits the impact of mutant channels on RTN neural function and breathing. For example, inhibition of Kcnq2 by wake-on neuromodulators may diminish genotype differences in channel function and so lessen the impact of loss of Kcnq2 function on baseline breathing in the dark/active state (Fig. 4D). Also, since RTN neurons that express Kcnq2[R201C] respond normally to serotonin (Fig. 6E, F), we speculate that inhibition of Kcnq2[R201C] by wake on neuromodulators offsets the gain-of-function effect on respiratory output in the dark/active state. Conversely, when arousal signaling is at its nadir during the light/inactive state, RTN neurons are subject to the full gain-of-function effect of Kcnq2[R201C] which manifests as a diminished ventilatory response to CO$_2$ (Fig. 5E). However, it remains to be determined whether breathing problems in DEE patients manifest in a sleep-wake or circadian dependent manner.

In sum, our work shows that disruption of Kcnq2 channel activity, either loss or gain of function in Phox2b-expressing neurons, leads to breathing abnormalities in vivo. Our work also shows that Phox2b-expressing ventral parafacial neurons primarily express Kcnq2 channels in the absence of other Kcnq channel isoforms. To the extent Phox2b-expressing ventral parafacial neurons are RTN neurons, these results suggest RTN neurons are vulnerable to Kcnq2 pathogenic variants. This provides an explanation for the presence of breathing

defects in patients with Kcnq2 GOF variants as compared to patients with Kcnq3 and Kcnq5 gain-of-function variants, which also increase Kcnq activity at resting membrane potentials.

A limitation of this work is that we were not able to measure arterial blood gases in conscious animals under baseline conditions. Therefore, we are not able to determine whether there are genotype differences in the regulation of blood gases or alveolar ventilation. We were also unable to obtain direct measures of metabolic activity while simultaneously measuring respiratory output, thus precluding normalization of minute ventilation with metabolism. Therefore, it remains unclear whether altered metabolic activity contributes to or masks respiratory disturbances in Kcnq2 cKO or Kcnq2 GOF mice, as described in other mouse models[40]. Lastly, although there is some evidence that ML252 delivered intraperitoneally can reach the brain[41], there is a scarcity of published studies describing the effects of ML252 on animal behavior so a more detailed dosage-response profile is needed. Therefore, that finding should be interpreted with caution.

## Methods
All procedures were performed in accordance with the National Institutes of Health and University of Connecticut Animal Care and Use Guidelines. Animals were housed in a 12:12 light/dark cycle with lights on at 7 a.m. or 9 p.m. for normal or inverted light/dark schedules (average ambient temperature 22 °C, average humidity 50%). Mice following the normal light:dark schedule was used for all in vitro experiments and for assessment of breathing during the light/inactive state measured between 9 a.m. and 5 p.m. To characterize breathing during the dark/active state, mice were acclimated for at least 5 days in housing with an inverted 12:12 light:dark schedule with lights on at 9 pm and experiments were performed between 11 am and 7 pm. Animals had unlimited access to normal chow and an enrichment hutch. We did not observe gender differences in minute ventilation in control mice ($T_{13} = 0.27$, $p = 0.79$), Kcnq2 cKO ($T_7 = 0.16$, $p = 0.87$) or Kcnq2 GOF mice ($T_7 = 0.54$, 0.61) so male and female mice were pooled for all comparisons.

### Kcnq2 encephalopathy models
*Kcnq2* encephalopathy can involve both loss and gain of *Kcnq* function variants[3,8]; therefore, to model this condition, we used the cre-lox system to conditionally delete *Kcnq2* or express *Kcnq2*$^{R201C/+}$ in *Phox2b* expressing neurons (Fig. 2A). Both mouse models were developed by the UConn Center for Mouse Genome Modification using recombineering as previously described[42]. Homozygous *Kcnq2* floxed (*Kcnq2*$^{f/f}$; 100% C57BL/6) mice were generated as described previously[43–45]. Briefly, exons 2–5 are flanked by LoxP sites. Upon expression of cre-recombinase, exons 2–5 are deleted, removing the vast majority of the transmembrane domains of the Kcnq2 channels. These animals were crossed with Phox2b-Cre BAC transgenic mice (B6(Cg)-Tg(Phox2b-cre)3Jke/J (JAX stock # 016223; for simplicity, these mice are defined as Phox2b::Cre BAC transgenic mice) to generate conditional *Kcnq2* knock-out mice (Kcnq2 cKO; B6(Cg)-Tg(Phox2b-cre)3Jke/J::*Kcnq2*$^{f/f}$).

To develop the conditional *Kcnq2*$^{R201C/+}$ (Kcnq2 GOF) model, we first retrieved ~6 kb of the *Kcnq2* genomic sequence containing exon 2 to 5 from RP24-389E11 into the PGKdta plasmid (Addgene plasmid no: 13440). A single Lox2372 site was then inserted into intron 3 approximately 200 bp 5'-upstream of exon 4. A DNA fragment containing the PGKneo cassette flanked by Frt sites followed by a LoxP site, exon 4 with the mutant R201C codon was placed in reverse orientation and opposite facing the Lox2372 site. Another LoxP was inserted in intron 4 of *Kcnq2*. The final *Kcnq2* R201C conditional knock-in targeting vector contains approximately 4 kb and 3 kb of 5' and 3' homology arms. This construct was then electroporated in in-housed D2 ES cells derived from F1(C57BL6x129sv) embryos. Electroporated ES cells were cultured in a medium containing G418 allowing for genomic DNA samples to be isolated from drug-resistant colonies. Colonies were initially screened by long-range PCR using primer pairs specific to the Lox sites and corresponding to *Kcnq2* sequences outside of the homology arms. Identity of positive colonies was further confirmed by long-range PCR and sequencing of the PCR product containing the *Kcnq2*$^{R201C}$ variant. Positive ES cells were then used to generate chimeric animals by aggregation with CD1 early morulae. Chimeric mice were then bred with CAG-Flpo mice (MMRCC stock no: 032247-UCD) to remove the PGKneo drug selectable cassette allowing for the generation of the final Kcnq2 R201C GOF mouse line containing the Kcnq2$^{R201C/+}$ allele (Fig. 2A).

To ensure that our mouse line works as designed, we crossed *Kcnq2*$^{R201C/+}$ mice with Hprt-cre mice (Jax stock #004302), a global Cre-expressing mouse line. Sequencing of genomic DNA confirmed that mice expressing cre-recombinase also express the inversion of mutant exon 4 with the R201C mutation in the sense transcriptional orientation (Supplementary Fig. 4). *Kcnq2*$^{R201C/+}$ mice were backcrossed onto a C57BL/6 background and bred with Phox2b::Cre BAC transgenic mice (100% C57BL/6) to generate a conditional *Kcnq2* gain-of-function model (Kcnq2 GOF; *Kcnq2*$^{R201C/+}$::B6(Cg)-Tg(Phox2b-cre)3Jke/J). Also, to facilitate cellular electrophysiological experiments, we generated a Kcnq2 GOF Phox2b reporter line by crossing Phox2b::Cre BAC transgenic mice with Ai14 Tdtomato$^{fl/fl}$ mice (100% C57BL/6) and the offspring of this was crossed with *Kcnq2*$^{R201C/+}$ (*Kcnq2*$^{R201C/+}$::B6(Cg)-Tg(Phox2b-cre)3Jke/J::*Tdt*$^{+/-}$). We note that recent work showed that generation of conditional knock-in mice using the FLEx switch system might lead to deletion of the endogenous exon due to unstable RNA structure[46]. Therefore, we looked for the presence of possible aberrant splicing in our *Kcnq2*$^{R201C/+}$ mice. We isolated cDNA from *Kcnq2*$^{R201C/+}$ mouse brains and PCR-amplified it using primers for *Kcnq2* exons 3 to 6. The resulting PCR product was inserted into pGEM-T easy vector. After transformation into NEB® 5-alpha Competent E. coli cells, colonies were screened on LB plates with ampicillin, IPTG, and X-gal. Selected colonies underwent Sanger sequencing using exon 3 primers, revealing wild-type or exon 4-lacking plasmids, indicating mis-splicing. Because of this, we decided to only compare *Phox2b::Cre:Kcnq2*$^{+/+}$ to *Phox2b::Cre:Kcnq2*$^{R201C/+}$ mice. For identifying the different mouse *Kcnq2* lines mice were genotyped using the following primers (*Kcnq2*$^{+/+}$, 172 bp; *Kcnq2*$^{R201C/+}$, 235 bp; *Kcnq2*$^{R201C/+}$, 263 bp): F: 5'GAGTTGGGTCTGGGGAGAGAG 3' and R: 5'GTGTCCAGGAGA-GACTGCCCAG 3' (Fig. 2B).

### Respiratory behavior
Respiratory activity was measured by whole-body plethysmograph (DSI/Buxco, St. Paul, MN) using a small animal chamber maintained at room temperature and ventilated at -1.15 L/min. Chamber temperature and humidity were continuously monitored and used to correct tidal volume breath by a breath basis. Mice (Kcnq2$^{+/+}$, Kcnq2 GOF, Kcnq2 cKO; P30-P50; mixed sex) were individually placed into the chamber and allowed -1 h to acclimate prior to the start of an experiment. Respiratory activity was recorded using Ponemah 5.32 software (DSI) for -30 min in room air followed by stepwise (10 min/condition) increases in $CO_2$ from 0 to 7% $CO_2$ against a hyperoxic background (balance $O_2$) to minimize peripheral chemoreceptor drive. In separate experiments, we characterized the ventilatory response to hypoxic stimuli: 10% $O_2$ (balance $N_2$). All plethysmography experiments were video recorded and sections of data containing behavior artifacts were excluded from analysis. Parameters of interest include respiratory frequency (breaths/minute), tidal volume ($V_T$, measured in mL; normalized to body weight and corrected to account for chamber temperature, humidity, and atmospheric pressure), and minute ventilation ($V_E$, mL/min/g). A 20 s period of relative quiescence after a minimum of 4–5 min of exposure to each condition was selected for analysis. In a separate analysis, 20 min sections of relative quiescence in room air were used for assessment of baseline respiratory pattern, apneas and sighs. An apneic event was defined as three or more missed breaths

that terminate when near-normal breathing frequency resumes. Sighs were identified based on their characteristic large amplitude (2× tidal volume). Note that spontaneous apneas and post-sigh apneic events were analyzed separately.

## Comprehensive lab animal monitoring (CLAMS)

Metabolic monitoring $O_2$ consumption ($VO_2$) and $CO_2$ production ($VCO_2$) was performed using comprehensive lab animal monitoring systems (CLAMS, Columbus Instruments). Adult mice (Kcnq2[+/+], Kcnq2 GOF; P30-P50) were individually housed on a 12:12 light:dark cycle in cages with a running wheel, bedding, and regular chow for one week before an experiment. Three days before metabolic assessment, each animal was placed in the CLAMS housing cage with metered water and waste collection. Mice were given two days to acclimate to the metabolic chamber; on the third day, all results were recorded for a continuous 24 h period (Oxymax v5.54). After data collection, results were exported and averaged per hour, only including times of no-wheel activity as assessed by an activity monitoring system within the CLAMS cage. Then, light and dark periods were determined and averaged per animal for statistical analysis. We focused our analysis on the respiratory exchange ratio (volume of CO2 produced/volume O2 consumed) and from this we estimated energy expenditure as heat (Kcal/h/kg) according to the manufacturer guidelines. Locomotor activity in the form of wheel running was measured separately. Both sexes were equally represented in the data set.

## Cellular Electrophysiology

Slices containing the RTN were prepared as previously described[20]. In brief, *Kcnq2[+/+]* (control) and Kcnq2 GOF mouse pups (P7-11 days postnatal, mixed sex) were anesthetized by administration of ketamine (375 mg/kg, I.P.) and xylazine (25 mg/kg; I.P.) and rapidly decapitated; brainstems were removed and transverse brainstem slices (250–300 μm) were cut using a microslicer (DSK 1500E; Dosaka) in ice-cold substituted Ringer solution containing the following (in mM): 260 sucrose, 3 KCl, 5 $MgCl_2$, 1 $CaCl_2$, 1.25 $NaH_2PO_4$, 26 $NaHCO_3$, 10 glucose, and 1 kynurenic acid. Slices were incubated for 30 min at 37 °C followed by room temperature in a normal Ringer's solution containing (in mM): 130 NaCl, 3 KCl, 2 $MgCl_2$, 2 $CaCl_2$, 1.25 $NaH_2PO_4$, 26 $NaHCO_3$, and 10 glucose. Both substituted and normal Ringer's solutions were bubbled with 95% $O_2$ and 5% $CO_2$ (pH = 7.3).

Individual slices containing the RTN were transferred to a recording chamber mounted on a fixed-stage microscope (Zeiss Axioskop FS) and perfused continuously (~2 mL/min) with a bath solution containing (in mM): 130 NaCl, 3 KCl, 2 $MgCl_2$, 2 $CaCl_2$, 1.25 $NaH_2PO_4$, 26 $NaHCO_3$, and 10 glucose (equilibrated with 5% $CO_2$; pH = 7.3). All recordings were made with an Axopatch 200B patch-clamp amplifier, digitized with a Digidata 1550B A/D converter and recorded using PClamp 11.0.3 software. Recordings were obtained at room temperature (~22 °C) with patch electrodes pulled from borosilicate glass capillaries (Harvard Apparatus, Molliston, MA) on a two-stage puller (P-97; Sutter Instrument, Novato, CA) to a DC resistance of 5–7 MΩ when filled with pipette solution. to minimize pipette capacitance and improve the signal-to-noise, electrode tips were coated with Sylgard 184 (Dow Corning, Midland, MI). Only one cell was recorded per slice.

Firing activity was measured in the tight seal (seal resistance >1 GΩ) cell-attached current-clamp (Vhold −60 mV) configuration using a pipette solution containing (in mM): 120 $KCH_3SO_4$, 4 NaCl, 1 $MgCl_2$, 0.5 $CaCl_2$, 10 HEPES, 10 EGTA, 3 Mg-ATP, 0.2% lucifer yellow, and 0.3 GTP-Tris (pH 7.2). Electrophysiological data was acquired and analyzed using Pclamp V11.0.3. software, and firing rate histograms were generated by integrating action potential discharge in 10–15 s bins using CED Spike 5.0 software. For each experiment, we introduce 10% $CO_2$ for at least 5 min or when a plateau of firing activity is achieved for at least 2 min. Resting membrane potential and input resistance

(measured during a −125 pA step) was measured within 30 seconds of gaining whole-cell access. Also, in whole-cell current-clamp, we characterized firing responses to depolarizing current steps (from +25 to +125 pA, Δ 20 pA, 1 s duration) from a holding potential of −65 mV. All whole-cell recordings had an access resistance <20 MΩ. A liquid junction potential of −10 mV was corrected off-line. After recording, slices were fixed with 2% PFA and stored @ 4 ˚C for up to one month before co-staining for Phox2b (1:100 dilution) [RRID:AB 10889846] and Lucifer Yellow (1:500 dilution) antibodies [RRID:AB_2536190] as described below.

## Immunohistochemistry

Weaned mice (Kcnq2[+/+], Kcnq2 GOF, Kcnq2 cKO; P20-50; mixed sex) were anesthetized with 3% isoflurane and transcardially perfused with 20 mL of room temperature phosphate-buffered saline (PBS, pH 7.4) followed by chilled 4% paraformaldehyde (pH 7.4 in 0.1 M PBS) by peristaltic pump. Brainstem sections were removed and allowed to post-fix for up to 4 h in 2% paraformaldehyde before transfer to 30% sucrose for 24–48 h at 4 °C. Tissue sections (75 μm thick; coronal plane) were collected using a Zeiss VT1000S vibratome. Slices were permeabilized by treating with a mixture of 0.5% Triton-X/10% normal horse serum/PBS for 2 h. Sections were then transferred to a 0.1% Triton-X/2% normal donkey serum/PBS mixture with the primary antibody (1:100, goat anti-mouse Phox2b (RRID:AB 10889846) and 1:250 rabbit anti-kcnq2 antibody (RRID: 2131689) overnight at room temperature. Slices were then washed in PBST (1x PBS, 0.02% Triton, 5 × 10 min) before incubating in blocking solution containing secondary antibodies (donkey anti-goat AlexaFluor647 for Phox2b and donkey anti-rabbit AlexaFluor488 for Kcnq2). After washing secondaries with PSBT (5 x 10min) and PBS (5x10min), slices were mounted on slides with ProLong Gold Antifade (Invitrogen, P36934) and DAPI (ThermoFisher). Slices were imaged using a Leica SP8 confocal microscope using 10× and 20× objectives, each image was Z compressed with full focus and max intensity using Fiji software. RTN neurons were identified by co-localization of Phox2b-immunoreactivity and DAPI along the ventral surface for ~600 μm medial to the trigeminal nucleus and from the rostral nucleus ambiguous (caudal border) to the facial nerve tracts (rostral border). The number of Phox2b+ cells was counted in Imange J and quantified manually across slices.

## Fluorescent in situ hybridization (FISH)

Animals (B6(Cg)-Tg(Phox2b-cre)3Jke/J::Ai14; P20-50; mixed sex) were anesthetized (ketamine, 75 mg/kg; xylazine, 5 mg/kg; IP) then transcardially perfused with 20 mL of room temperature phosphate-buffered saline (PBS, pH 7.4) followed by chilled 2% paraformaldehyde (pH 7.4 in 0.1 M PBS) by peristaltic pump. After fixation, brains were removed, frozen on a stage with 100% ETOH and dry ice then encased in mounting media and stored at −80 C until further processing. Coronal slices (20 μm) were cryosectioned, mounted on super frost slides and stained according to manufacturer instructions (RNAscope Multiplex Fluorescent Assay, Advanced Cell Diagnostics (ACD), Hayward, CA; RRID:SCR_012481). The probes used in our study were designed and validated by ACD and include: *Kcnq2* (Catalog # 407861) and *Kcnq3* (catalog # 444261). Multiplex in situ hybridization was combined with immunohistochemical detection of mTomato as described[14]. For this, slides were processed as described above, then rinsed for 10 min in blocking buffer (10% horse serum, 0.1% triton in 100 mM Tris buffer) and incubated in blocking buffer containing primary antibody for mTomato for 1 h (4 °C, 1:400 rabbit α-dsRed, Takara Biosciences, AB_10013483). Sections were rinsed 2 ×2 min in Tris buffer, incubated for 30 minutes in Tris buffer with secondary antibody (1:500, donkey α-rabbit-Cy3, AB_2307443; both Jackson Immunoresearch), rinsed and allowed to air dry. Slides were covered with Prolong Gold with DAPI Anti-fade mounting medium (ThermoFisher

Scientific, Catalog#P36935). Confocal images of FISH experiments were obtained using a Leica TSC SP8 and LAS X acquisition software at 1024 × 1024 and 512 × 512 resolution, with a digital zoom of 1 and a minimum total Z stack of 10 μm. Files containing image stacks were loaded into ImageJ v2.0.0 for analysis and all Z-stacks were merged to maximum intensity and brightness and contrast were adjusted to minimize background and channel saturation. To reduce noise, a median filter was applied to each channel (1 pixel). Cells were counted on each channel individually using the Tdt channel to distinguish an individual cell from background. Any cell that was partially out of frame was not included in the analysis. Cells are considered positive for a transcript if the tdT labelled soma showed five or more individual puncta.

### Single-cell isolation and qRT-PCR

An enriched population of Phox2b expressing neurons from the RTN region were obtained from a total of 3 adult B6(Cg)-Tg(Phox2b-cre) 3Jke/J::Ai14 mice (>P21) as previously described[22]. In brief, animals were euthanized under anesthesia (3% isoflurane) and brainstem slices were prepared using a vibratome in ice cold, high sucrose slicing solution containing (in mM): 87 NaCl, 75 sucrose, 25 glucose, 25 NaHCO3, 1.25 NaH2PO4, 2.5 KCl, 7.5 MgCl2, 0.5 mM CaCl2, and 5 l-ascorbic acid. Slicing solution was equilibrated with a 5% CO2−95% O2 gas mixture. Transverse slices (300 μm thick) were prepared and transferred to a glass Petri dish containing ice-cold dissociation solution composed of (in mM): 185 sucrose, 10 glucose, 30 Na2SO4, 2 K2SO4, 10 HEPES, 0.5 CaCl2, 6 MgCl2, 5 l-ascorbic acid, pH 7.4, 320 mOsm. Using a plastic transfer pipette and scalpel (15-blade), the RTN was isolated and manually separated into sterile microcentrifuge tubes. The tissue chunks were then warmed to 34 °C for 10 min followed by trituration using a 25G and 30G needle sequentially, attached to a 3 mL syringe. Samples were triturated for an average of 5 min. The samples were placed back on ice and filtered through a 30-micron filter (Miltenyi Biotech) into a round bottom polystyrene tube for FACS.

### Florescence-activated cell sorting (FACS)

A single cell suspension of Phox2b+ ventral parafacial cells was sorted using a BD FACSAria II Cell Sorter (UConn COR2E Facility, Storrs, CT) equipped with 407 nm, 488 nm, and 607 nm excitation lasers using FACSDiva 8.0 software. Five minutes before sorting, 5 μL of 100 ng/mL DAPI was added to each sample. Cells were gated based on scatter (forward and side), for singlets, and for the absence of DAPI. Finally, cells were gated by TdTomato fluorescence and sorted by four-way purity into a sterile 96-well plate containing 5 μL of sterile PBS per sample. Between 100 and 500 TdTomato cells were sorted per sample in any experiment and were processed immediately following FACS.

### Pooled cell qRT-PCR

A lysis reaction followed by reverse transcription was performed using the kit Taqman Gene Expression Cells-to-CT Kit (ThermoFisher) with 'Lysis Solution' followed by the 'Stop Solution' at room temperature, and then a reverse transcription with the 'RT Buffer', 'RT Enzyme Mix', and lysed RNA at 37 °C for an hour. Following reverse transcription, cDNA was pre-amplified by adding 2 μL of cDNA to 8 μL of preamp master mix [5 μL TaKaRa premix Taq polymerase (Clontech), 2.5 μL 0.2X Taqman pooled probe, 0.5 μL H2O] and thermocycled at 95 °C for 3 min, 55 °C for 2 min, 72 °C for 2 min, then 95 °C for 15 s, 60 °C for 2 min, 72 °C for 2 min for 16−20 cycles, and then a final 10 °C hold. Amplified cDNA was then diluted 2:100 in RNase-free H2O. Each qPCR assay contained the following reagents: 0.5 uL 20X Taqman probe (*Kcnq1* #Mm00434638_m1, *Kcnq2* #Mm00440080_m1, *Kcnq3* #Mm00548884_m1, *Kcnq4* #Mm01185500_m1, *Kcnq5* #Mm01226041_m1, *Gapdh* #Mm99999915_g1), *Rbfox3* #Mm01248781_ml, *Olig1* #Mm00497537_sl, *Gfap* #Mm01253033_ml,

*Aldh1l1* #Mm03048957_ml), 2.5 μL RNase-free H2O, 5 μL Gene Expression Master Mix or Fast Advanced Master Mix (ThermoFisher), and 2 L diluted pre-amplified cDNA. qPCR reactions were performed in triplicate for each Taqman assay of interest on a QuantStudio 3 Real-Time PCR Machine (ThermoFisher) and analyzed on QuantStudio Design & Analysis Software v1.5.1. Three technical replicates were averaged to create one raw Ct values. Any assay that did not give a discrete Ct value was given a Ct value of 40 for analysis. *Gapdh* was used as a sample-dependent internal control. ΔCt values were calculated by subtracting *Gapdh* signal from any gene of interest and averaging across triplicates.

### Statistics

Power analysis was used to determine sample size, all data sets were tested for normality using the Anderson-Darlington, D'Agostino & Pearson, and Shapiro-Wilk tests, and outlier data points were identified by the Grubbs test and excluded from analysis. Normally distributed data was analyzed using paired or unpaired T-test, one-way or two-way ANOVA followed by Tukey's multiple comparison test. Data sets that failed all three normality tests above are considered non-normally distributed and analyzed using the Mann-Whitney test followed by the Holm-Sidak multiple comparison test. Also, the non-parametric analysis of covariance was used to compare slopes of linear regressions. The specific test used for each comparison is reported in the figure legend and all relevant values used for statistical analysis are included in the results section. Summary data are plotted as mean ± SEM along with individual data points, differences between means are considered significant when $p < 0.05$.

### Reporting summary

Further information on research design is available in the Nature Portfolio Reporting Summary linked to this article.

## Data availability

The authors declare that all source data supporting the claims in this study are provided as a Source Data file. The raw scRNAseq data supporting findings in this study have been deposited in the GEO Repository with the primary accession code (GSE153172). Additional raw data that support this study are available from the corresponding authors upon request. Source data are provided with this paper.

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

## Acknowledgements

The authors thank Drs. Heun Soh and Nissi Varghese for testing the potential mis-splicing in Kcnq2^R201C/+ mice. This work was supported by the following National Institutes of Health Grants: HL104101

(DKM), NS134132 (DKM), HL137094 (DKM and AVT), NS101596 (AVT) and F31 HL156470 (JS-P).

## Author contributions

J.S.-P.: Designed experiments, generated data, analyzed results, edited the manuscript, and approved the final manuscript. C.M.C.: Designed experiments, generated data, analyzed results, edited the manuscript, and approved the final manuscript. CRS: generated data, analyzed results, edited the manuscript, and approved the final manuscript. S.B.M.: Designed experiments, edited the manuscript and approved the final manuscript. J.L.C.: Designed experiments, edited the manuscript and approved the final manuscript. A.V.T.: Designed experiments, drafted the manuscript and approved the final manuscript. D.K.M.: Designed experiments, drafted the manuscript and approved the final manuscript.

## Competing interests

The authors declare no competing interests.
