## [Peer Review File · Nature Communications]

Phox2b-expressing neurons contribute to breathing problems in Kcnq2 loss- and gain-of-function encephalopathy modelsREVIEWER COMMENTS

Reviewer #1 (Remarks to the Author):

Selected KCNQ2 channels gain or loss of function mutations cause respiratory disturbances among many other issues. The authors propose that the retrotrapezoid nucleus (RTN), a 700-neuron lower brainstem nucleus is a key contributor to the breathing abnormalities caused by KCNQ2 gain or loss of function. The given rationale is that a) the RTN is important for the metabolic regulation of breathing (arterial PCO₂ stability) and b) it expresses high levels of KCNQ2 channels, a fact that the authors establish convincingly at the outset of this study.

The authors uncovered the existence of clear anomalies in resting discharge rate, excitability or CO₂ - response of RTN neurons in neonatal slices from mice that express loss or gain of function KCNQ2 channels. They also show that the deleterious effect of the gain of function mutation can be negated by a KCNQ2 blocker or the presence of serotonin, a wake-promoting transmitter that works in part by inhibiting KCNQ2 channels. This interesting fact can be viewed as consistent with their observation that the breathing deficit of GOF mice is more pronounced while they are inactive (presumably mostly sleeping) although we still do not know for a fact that the release of serotonin in the RTN is state-dependent.

The electrophysiological evidence from slices is convincing and the authors correctly suggest that these anomalies could indeed contribute to the breathing problems, including sleep apnea, experienced by rodents or human exhibiting these mutations. However multiple brain regions from the cortex down to the spinal cord are, like the RTN, either part of, or regulate the brainstem respiratory pattern generator and likely express KCNQ2 channels.

The authors' case, namely that the breathing disorder is caused by the presence of mutated KCNQ2 channels in RTN neurons specifically or primarily, is largely based on genetic crosses with mice expressing the cre-recombinase under the control of the Phox2b gene. True, RTN neurons are among the CNS population of Phox2b- expressing neurons but this nucleus probably comprises less than 1% of all Phox2b-dependent lower brainstem and autonomic neurons. The vast majority of the Phox2b- expressing neurons are located in regions that are also critically important to the control of breathing (nucleus solitary tract, dorsolateral pons, catecholaminergic neurons including the locus coeruleus, carotid body and their primary afferents, etc.). For this reason, the authors should be much more cautious a priori in attributing the whole animal changes in breathing to the particular RTN anomalies they have uncovered, especially since KCNQ2 was detected in only 60% of the RTN neurons. The authors case that the presence of mutated KCNQ2 channels in RTN underlies some or all of the respiratory abnormalities would of course be far more convincing if they had targeted the mutated channels selectively to the RTN.

Finally, the characterization of the mutant mice should be expanded somewhat to include measures of Ve/VCO₂ and arterial PCO₂ which could provide valuable information on the performance of the RTN by determining whether the mice hypo or hyperventilate at rest.

In summary, this is a well-executed but somewhat limited and overinterpreted study showing that pathogenic mutations of the KCNQ2 channels (gain or loss of function) change the excitability of the RTN neurons in mice in a manner that could theoretically contribute to the observed changes in the central hypercapnic response of the mutant animals. Causality is far from proven however because the mutated channels were presumably expressed in every Phox2b-dependent neuron. In this reviewer's opinion, the authors should be more conservative in their interpretations unless they can come up with a way to test what happens when the mutated KCNQ2 channels are expressed exclusively by RTN neurons.

Major points

- Please say something about what is known re. Kcnq2 expression in other Phox2b-expressing cells, and how that can affect interpretation of the behavioral results.
- What is the expectation for the KCNQ2 cKO mice in the dark, active-state conditions? and why was

that not tested? Based on the same speculation advanced for the GOF mutant (i.e., that there is no difference under these conditions because the channels are usually inhibited), one might also predict no difference for those cKO mice vs. wt mice in the dark, active-state conditions as well.

- Is the lack of effect in dark, active-state conditions due to this channel modulation by wake-on transmitters, as speculated here, or instead due to a lesser role for RTN more generally in that state? How can those two possibilities be disambiguated? This should be discussed.

- Also, on this point, are the respiratory problems in DEE patients manifest only in the inactive state?

- The cell selection process for defining a recorded cell as an RTN neuron could be more clear. It is stated that "Chemosensitive RTN neurons were identified based on location within the ventral parafacial region (Fig. 5A), a minimum 0.5 Hz excitatory response to 10% CO₂, and in some cases PHOX2B immunoreactivity or Phox2b-dependent tdT expression (Fig. 5A)" but then in the next sentence "However, approximately half of RTN neurons in slices from KCNQ2 GOF mice were not spontaneously active under control conditions and those that were spontaneously active were less active under control conditions (0.7 ± 0.2 Hz, $p = 0.014$) and showed a blunted response to 10% CO₂ compared to RTN neurons in control tissue (1.0 ± 0.1 Hz, $p = 0.0003$) (Figs. 5C-E)." So, at least some of those GOF neurons did not meet the required criterion for RTN identification (delta FR of 0.5 Hz). Then, later, "Neurons that did not show a minimum CO₂ response (even after adjusting baseline activity) were excluded from further analysis" but it is not clear what this minimum CO₂ response threshold is for inclusion.

- Abstract Lines 52-54: "We find that respiratory neurons in the RTN that express the transcription factor PHOX2B also preferentially express KCNQ2 in the absence of other KCNQ isoforms, thus explaining why disruption of KCNQ2 but not other channel isoforms results in breathing problems". The first part of the statement is on solid grounds but the second one (thus explaining why etc.) seems far-fetched. The authors do indeed show that KCNQ2 mutations alter the properties of RTN neurons in tissue slices in a manner that could potentially account for the changes in whole body chemoreflex but they provide no evidence that RTN neurons are the only Phox2b -dependent neurons that express KCNQ2 and regulate breathing. Also, are the authors sure that neurons that express KCNQ2 along with other isoforms are unaffected by these KCNQ2 mutations? The fact that mutations of KCNQ isoforms other than KCNQ2 do not produce the same phenotype does not seem to provide a solid argument against the latter possibility.

- Lines 84-86: these statements should be less assertive. There is still no direct evidence that the RTN is absent in CCHS patients. This is a hypothesis based on a mouse genetic model (primarily Phox2b27ala and related Phox2b27alacki). Also, the Dubreuil paper (ref 14) actually shows that the pan-Phox2b (Phox2b27ala) mutation is lethal in mice (as it would be in humans without respiration assistance) and the same authors have later shown that a presumably complete and much more selective genetic lesion of the RTN (Phox2b27alacki) does not produce life threatening sleep apnea (since the mice uniformly survive); in fact this latter RTN hypoplasia model exhibits an incomplete (60%) reduction of the hypercapnic chemoreflex in adulthood (Ramanantsoa et al.). Finally, in adult rodents, virtually total lesions of the RTN produce hypoventilation but do not produce sleep apnea.

- Results section entitled: "chemosensitive RTN neurons preferentially express Kcnq2 transcript" The authors show that only 57-66% of RTN neurons (identified as positive for Slc17a6, Phox2b, Nmb, mRNA and the defining proton sensors Gpr4 and Kcnk5 mRNA) express KCNQ2 transcripts. Other KCNQ subtypes could not be detected. The KCNQ3 isoform was generally undetectable. The authors interpret these data as suggesting that KCNQ2 is selectively expressed by RTN neurons and propose that the channels are homomeric in this cell type, unlike in other neurons. The demonstration is not totally convincing; a quarter of the KCNQ2-positive RTN neurons also contained detectable levels of KCNQ3 mRNA (lines 512). KCNQ3 mRNA may simply be present at a lower level than KCNQ2 or may be less effectively detected by hybridization probes. It would be nice to know whether the existence of a high KCNQ2 / KCNQ3 mRNA ratio is a unique characteristic of RTN neurons or a feature present in

most other neurons. For example, was KCNQ2 mRNA present at about the same level as KCNQ3 mRNA in the facial motor neurons or was it much higher as was the case in RTN neurons? In short the evidence that "homomeric KCNQ2 channels are preferentially expressed by RTN neurons" (line 266) is not entirely convincing.

- Results section entitled: KCNQ2 loss- and gain-of-function variants differentially affect baseline breathing and the central chemoreflex.

Lines 145-146, Figure 3 and Suppl Figure 2. The authors focus on the respiratory quotient (V_{CO2}/V_{O2}) which is basically a measure of which fuel (lipids vs glucose) is preferentially consumed for energy production as opposed to a direct measure of metabolic activity. It seems that reporting the V_e/V_{CO2} ratio might have been more informative in the context of this study. Resting V_{CO2} and V_e/V_{CO2} ratio would help determine whether the mutations cause hyper or hypoventilation at rest, a diagnosis ideally to be confirmed by direct measurements of the arterial PCO_2 . For example, although resting V_e is no different between wildtype and GOF mice (Figs 3 and 4), the GOF mutant mice could very well hypoventilate at rest (i.e. when no CO_2 is added) meaning that V_e might be the same as that of the controls only because of a rise in steady state $PaCO_2$. This is typically what happens after moderate size RTN lesions in rodents. Such a result would be consistent with a lower activity of RTN neurons relative to the prevalent $PaCO_2$ in the GOF mutants (Figure 5B) and would strengthen the authors' notion that the breathing defect originates in part from the presence of mutated KCNQ2 channels in RTN neurons.

Minor

- The text should probably state explicitly that the same light, inactive-state data from GOF neurons in Fig. 3 is re-plotted again with the corresponding dark, active-state data from GOF neurons in Fig. 4.
- Running title. Please define somewhere what you mean exactly by chemoreception and consider modifying this title or not, accordingly. Regrettably, the word chemoreception is used in the literature in reference to anything from molecules (proton receptors) to whole animal breathing reflex. The present data indicates that the RTN response to CO_2 is sensitive to the channelopathies but is the effect of protons on membrane conductance affected? What changes seem to be cell excitability and the ability to transduce pH stimuli into increased firing. You can define the whole RTN cellular response to CO_2 as chemoreception if you wish but it better be defined.
- The discussion of channel subunit stoichiometry seems overly specified, especially since the relative expression, assembly affinity, etc. of the different variant and isoform subunits is unknown. The authors do qualify that their numbers apply if assuming a binomial distribution, but their point is valid and understandable even without the pseudo-quantification.
- There is no description of how plethysmography experiments were performed at different times of day. In fact, the text says "All experiments were performed between 9 a.m. and 6 p.m. to control for potential circadian effects." So, did someone come in late at night? Were the mice put in rooms with inverted light:dark schedule?
- What is the Phox2b-Cre line? The text designation as Phox2b⁺/Cre makes it seem like a knock-in mouse (i.e., Cre at the Phox2b locus) but the mouse line classically used from Jax is a BAC transgenic? Please describe the line and annotate properly if a BAC transgenic.
- Fig. 1 legend is not right. Also, the text says that Kcnq3 expression in nearby facial motoneurons is "not shown" but there appear to be some neurons in 7N of Fig. 1B.
- Fig. 5 legend, "B, traces of firing rate and segments of membrane potential from chemosensitive RTN ..." should be revised since "membrane potential" is not measured in these cell-attached recordings.
- Since this is the first description of these GOF mice, and other PIs might be interested, please provide supplemental data showing the mis-splicing that is referenced in the text (or a more complete description of how that was determined).
- p. 12 leading to "higher" resting membrane potential? means more depolarized?
- Please show locations of forward and reverse primers for assessing the GOF mouse. It would also be helpful if the schematic in Fig. 2A also showed the wt allele.

- Supplemental figure 1A/B. Please check the captions (CGG)

Reviewer #2 (Remarks to the Author):

This is an interesting manuscript demonstrating a pathological role of gain of function mutation of KCNQ2(R201C) using conditional knock-in mice that selectively express at RNT neurons. This paper is significant because this determined the cause of breathing problem in patients with KCNQ2(R201C) mutation. Authors demonstrated basic characterization of Phox2b-Cre mediated KCNQ2(R201C) knock-in mice. Followed by functional characterization of respiratory regulation in these knock-in mice. These knock-in mice had normal basal respiration, while had less CO₂ induced hyperventilation at light/inactive state. These behavioral differences were confirmed in brain slice experiments. All experiments are logically connected, and results are convincing and they are in good quality. Conclusion and discussion of this paper is valid and supported by the robust experimental results with appropriate analyses.

Major comment. One thing that this reviewer felt missing in this manuscript is that whether this pathological phenotype in KCNQ2(R201C) mice can be treated by ML 252 in whole animal. The authors already use ML252 in their brain slice experiments in Fig. 6. This proposed experiment will elucidate whether KCNQ2(R201C) is the only cause of this phenotype. If they can demonstrate ML 252 can be used to treat GOF mice, this manuscript will have a greater significance on potential therapy.

Specific comments. Some sections of the manuscript were unclear and need revision.

- 1) Fig. 1. Legend. Explanation for panel C (B) is missing. Report total cell counts for panel C.
- 2) Fig. 2C report cell count for each pie chart.
- 3) Fig. 3 Labeling Bi, Bii, Biii, Ci, Cii, Ciii. I am not sure this labeling system is adequate for journal style. Please check. Also, report n for each genotype.
- 4) Fig. 4 Labeling problems. Same with above.
- 5) Fig. 5E. This panel does not seem to represent the results shown in panel F. It looks 5HT has less effects in GOF mice.
- 6) Page 7, line 13-14, & 22-23. Redundant.
- 7) P8. First sentence. Logic unclear need clarification.
- 8) P9. ML252, some papers describe as KCNQ2 selective blocker. Why it is describe pan-KCNQ channel blocker in this paper? Need clarification.
- 9) P11. Discussion regarding haploinsufficiency is very difficult to follow.
- 10) P11. Discussion about retigabine and serotonin. Which 5-HT receptors are expressed in RNT neurons? The logic is very difficult to follow. If 5-HT can cause KCNQ2 current suppression, effect of retigabine can be canceled. Related with this, but as mentioned above, it will be very interesting to know whether ML 252 can restore CO₂ regulation in GOF whole animal.

Reviewer #3 (Remarks to the Author):

In humans, KCNQ2 mutations result in severe developmental and epileptic encephalopathy (DEE) and breathing disturbances that include profound hypoventilation and apnea. How KCNQ2 mutations and these respiratory phenotypes are linked is not known. Soto-Perez et al. hypothesize that alterations in KCNQ2 expressed in Phox2B+ parafacial respiratory neurons underlie the respiratory dysfunction observed in human patients with KCNQ2 mutations. The authors first use transcriptomics, FISH, and IHC to show that KNCQ2 is the sole KCNQ isoform expressed in a majority of Phox2B neurons. The authors then find that baseline breathing and CO₂ ventilatory responses are differentially affected in a state-dependent manner in loss and gain of function KCNQ2 transgenic mouse models where the perturbation is limited to Phox2B-expressing neurons. Finally, the authors use single cell

electrophysiology to demonstrate that the GOF KCNQ2 variant alters intrinsic properties of Phox2B neurons in ways that are consistent with the behavioral effects. The authors conclude that alterations in KCNQ2, particularly gain of function mutations, in Phox2B neurons play a role in regulation of breathing that may underlie the respiratory phenotypes seen in humans with KCNQ2 mutations.

The authors extend their previous work on the effects of pharmacological blockade of KCNQ on breathing and address a major gap in our understanding of a clinically relevant channelopathy. The experiments shed light on the specific role of KCNQ2 in the neural control of breathing using multiple approaches that rigorously characterize the effects of a highly specific perturbation from single neuron functional properties to in vivo behavior. The use of both loss and gain of function mouse models that are conditionally expressed in the Phox2B population powerfully exposes state-dependent roles of KCNQ2 in these neurons that provide a link between genotype and phenotype. The in vivo breathing assays and single cell electrophysiology in vitro are robust and the findings are consistent with both their previous work and the clinical presentation in humans. The data are high quality and are clearly presented in text and figures. Analysis and statistics are appropriate; although, a few minor changes (see below) would strengthen the manuscript. The data support their conclusions, and the manuscript will be of interest to basic neuroscientists interested in K⁺ channels and neural control of breathing and may also be translationally relevant for those studying channelopathies and conditions with respiratory phenotypes.

No major issues noted.

Minor comments:

Statistics. A statistics paragraph in methods would be helpful to understand some of the statistical details throughout the manuscript. Specific issues are listed below.

- The rationale behind the use of the different post-hoc tests for the various ANOVAs should be described. Tukey, Sidak, and Dunnett's are all reported, and it is not obvious why different post-hoc tests were chosen.

- A statement about how the data were determined to be normally distributed should be provided to justify the exclusive use of parametric tests.

- Both F statistic and p-value should be reported wherever an ANOVA is run.

- Details of the group data figures, bar, scatter, and box plots, particularly whisker values and what error bars represent (sd or sem), should be reported.

- What is the rationale behind using box plots for some graphs and mean \pm sd scatter plots for other panels given that all data are, I assume, normally distributed?

- A 2 way repeated measures ANOVA is reported for Fig 6B experiments, but I do not think this test is appropriate for the study design, which involves two independent groups that do not undergo all "treatments."

p. 5, line 117. Levels of other Kcnq isoforms in transcriptomic analysis are stated "not shown", but Fig 1A appears to show the data and should be referenced. Figure 1 legend also omits a description of Figure 1A.

p. 6, lines 142-150. Authors note the lower than expected frequencies of mutants and conclude that both cKO and GOF may be associated with embryonic lethality. The expected Mendelian frequencies for both genotypes (50% for cKO and 25% for GOF?) and number of litters collected for this analysis should be reported. The possible causes of embryonic lethality should be discussed given the relatively mild phenotype of the animals that survived and the fact that breathing is not necessary for survival in utero. Is there any evidence for perinatal, rather than embryonic, lethality?

p. 6, line 152-3. Based on this sentence, it seems like the analysis of baseline breathing in cKO is limited to the light/inactive state, yet the specificity of the difference in baseline breathing to the light/inactive phase (or that it was only measured during light) is not apparent when this result is

presented in abstract, introduction, Figure 3, and discussion. If analysis was restricted to light/inactive, this should be made explicit throughout. Given the increase in serotonin during wakefulness and its effects on KCNQ, it would be interesting to note the effects, if any, of the cKO on baseline breathing and CO₂ response during the dark/awake state and on sighs and apneas for both states.

p. 8, line 199-203. Is the firing rate for 10% CO₂, the change in firing rate? The values as reported are very similar and figure 5D shows a delta CO₂. This should be clarified in the text and figure legend.

p. 11, line 266. On p. 12, line 289-294, the role of serotonin in perhaps limiting the GOF effect to light state is discussed. A complementary discussion on serotonin and cKO would be similarly enlightening. From the Hawryluk et al. 2012 paper, XE991 had an effect on baseline breathing in anesthetized, but not awake rats, though KCNQ blockade blunted 5HT responses in both states. It seems that discussing how a blunted serotonin effect in cKO might be compensated for in awake states would be interesting.

p. 15, lines 374-375. The authors note that all experiments were performed between 9 am and 6 pm and earlier note that animals were housed with a 12:12 light/dark cycle. The time of the switch should be reported to make this statement meaningful and illuminate when the dark/active experiments were performed.

p. 21, line 515. Figure 2 legend title is unclear.

p. 22, line 537. "effects" should be "affects"

Figure 1A. Using a bar graph with n=3 data points that are almost all overlapping is a bit unclear. It might be clearer to just plot the individual data points and a scatter plot with mean +/- sd.

Figure 2c. Scale bar value missing.

Figure 4Bi, legend, "Kcnq2" should be capitalized.

Figure 4Ci, ii, "bal.O2" in Ci and "balance O2" in Cii should be made consistent.

Figure 5A right, scale bar value missing.

Response to Reviewer Concerns:

Summary: We thank the reviewers and editor for their thoughtful handling of this manuscript. We have addressed all reviewer concerns to the best of our ability and modified the text and figures accordingly. In particular, we performed additional experiments to more thoroughly characterize Kcnq2 and Kcnq3 transcript expression in brainstem Phox2b-expressing neural populations. Consistent with our hypothesis, we found that RTN neurons preferentially express Kcnq2, whereas the majority of Phox2b neurons in other brainstem regions co-express both Kcnq2 and Kcnq3 transcript. We also performed additional experiments to characterize respiratory activity of Kcnq2 cKO mice during the dark/active state. We found that Kcnq2 cKO mice exhibit breathing problems during the light/inactive state but not during dark/active conditions. This new finding complements our initial observations, and together suggests behavioral impacts of Kcnq2 channel mutations are circadian dependent. Unfortunately, efforts to improve breathing in Kcnq2 GOF mice by systemic injection of a pan-Kcnq2 channel blocker (as suggested by reviewer 2) did not yield believable results and so were not included in the revised submission (please see below for more details). We also edited the text and figures for improved clarity and expanded the discussion to speculation regarding the apparent light/inactive state dependent nature of Kcnq2 loss and gain of function effects on respiratory behavior.

Sex as a biological variable was compared. We report in the text that no gender differences in minute ventilation were observed between control, Kcnq2 cKO or Kcnq2 GOF mice so data from male and female mice were pooled for all subsequent analysis.

Please see below our point-by-point responses to noted concerns.

Reviewer #1 Comments:

Selected KCNQ2 channels gain or loss of function mutations cause respiratory disturbances among many other issues. The authors propose that the retrotrapezoid nucleus (RTN), a 700-neuron lower brainstem nucleus is a key contributor to the breathing abnormalities caused by KCNQ2 gain or loss of function. The given rationale is that a) the RTN is important for the metabolic regulation of breathing (arterial PCO₂ stability) and b) it expresses high levels of KCNQ2 channels, a fact that the authors establish convincingly at the outset of this study.

The authors uncovered the existence of clear anomalies in resting discharge rate, excitability or CO₂ -response of RTN neurons in neonatal slices from mice that express loss or gain of function KCNQ2 channels. They also show that the deleterious effect of the gain of function mutation can be negated by a KCNQ2 blocker or the presence of serotonin, a wake-promoting transmitter that works in part by inhibiting KCNQ2 channels. This interesting fact can be viewed as consistent with their observation that the breathing deficit of GOF mice is more pronounced while they are inactive (presumably mostly sleeping) although we still do not know for a fact that the release of serotonin in the RTN is state-dependent.

The electrophysiological evidence from slices is convincing and the authors correctly suggest that these anomalies could indeed contribute to the breathing problems, including sleep apnea, experienced by rodents or human exhibiting these mutations. However multiple brain regions from the cortex down to the spinal cord are, like the RTN, either part of, or regulate the brainstem respiratory pattern generator and likely express KCNQ2 channels.

The authors' case, namely that the breathing disorder is caused by the presence of mutated KCNQ2 channels in RTN neurons specifically or primarily, is largely based on genetic crosses with mice expressing the cre-recombinase under the control of the Phox2b gene. True, RTN neurons are among the CNS population of Phox2b-expressing neurons but this nucleus probably comprises less than 1% of all Phox2b-dependent lower brainstem and autonomic neurons. The vast majority of the Phox2b-expressing neurons are located in regions that are also critically important to the control of breathing (nucleus solitary tract, dorsolateral pons, catecholaminergic neurons including the locus coeruleus, carotid body and their primary afferents, etc.). For this reason, the authors should be much more cautious a priori in attributing the whole animal changes in breathing to the particular RTN anomalies they have uncovered, especially since KCNQ2 was detected in only 60% of the RTN neurons. The authors case that the presence of mutated KCNQ2 channels in RTN underlies some or all of the respiratory abnormalities would of course be far more convincing if they had targeted the mutated channels selectively to the RTN.

Finally, the characterization of the mutant mice should be expanded somewhat to include measures of Ve/VCO2 and arterial PCO2 which could provide valuable information on the performance of the RTN by determining whether the mice hypo or hyperventilate at rest.

In summary, this is a well-executed but somewhat limited and overinterpreted study showing that pathogenic mutations of the KCNQ2 channels (gain or loss of function) change the excitability of the RTN neurons in mice in a manner that could theoretically contribute to the observed changes in the central hypercapnic response of the mutant animals. Causality is far from proven however because the mutated channels were presumably expressed in every Phox2b-dependent neuron. In this reviewer's opinion, the authors should be more conservative in their interpretations unless they can come up with a way to test what happens when the mutated KCNQ2 channels are expressed exclusively by RTN neurons.

We thank the reviewer for their time and thoughtful suggestions.

Major points

1) Please say something about what is known re. Kcnq2 expression in other Phox2b-expressing cells, and how that can affect interpretation of the behavioral results.

Good suggestion. The discussion notes that Kcnq2 channels are expressed together with other channel isoforms in respiratory centers like the pre-böttinger complex and hypoglossal motor neurons; however, little is known regarding expression of this channel in Phox2b expressing neural populations. Therefore, we chose to address the reviewers concern experimentally by performing additional fluorescent *in situ* hybridization experiments to characterize expression of Kcnq2 and Kcnq3 transcript in Phox2b-expressing neural populations in the facial motor nucleus (7N), locus coeruleus (LC) and nucleus tractus solitarius (NTS). Note the LC and NTS are important respiratory centers. Perhaps not surprising, we detected abundant expression of Kcnq2 and Kcnq3 in Phox2b neurons in the 7N, LC and NTS. These new results have been added to Figure 1.

The co-expression of Kcnq2 together with Kcnq3 is important because these channels preferentially heteromerize when expressed in the same cell (PMID: 9836639; PMID: 10684873), and this will effectively dilute the impact of mutant Kcnq2 channels on heteromeric channel conductance (PMID: 25740509; PMID: 35780567). Considering RTN neurons predominantly express Kcnq2 channels with limited co-expression of KCNQ3 (Fig. 1A), unlike NTS, LC, and 7N neurons (Figs. 1B-F), we anticipate the effects of Kcnq2R201C to be more pronounced in RTN neurons compared to other Phox2b neurons. We have added this point to the discussion.

2) What is the expectation for the KCNQ2 cKO mice in the dark, active-state conditions? and why was that not tested? Based on the same speculation advanced for the GOF mutant (i.e., that there is no difference under these conditions because the channels are usually inhibited), one might also predict no difference for those cKO mice vs. wt mice in the dark, active-state conditions as well.

Good suggestion. We performed this experiment and as anticipated by the reviewer, we found that high baseline breathing observed in Kcnq2 cKO mice under light/inactive state was no longer different from control mice in the active/dark state. This outcome is consistent with the possibility that inhibition of Kcnq2 channels by wake-on neuromodulators limits effects of Kcnq2 loss or gain of function on respiratory activity. These new results have been incorporated into the text and displayed as a new Figure 4.

3) Is the lack of effect in dark, active-state conditions due to this channel modulation by wake-on transmitters, as speculated here, or instead due to a lesser role for RTN more generally in that state? How can those two possibilities be disambiguated? This should be discussed.

Good point. There is some evidence suggesting RTN chemoreceptors are particularly important for maintaining breathing during sleep (specifically nrem sleep) compared to during wakefulness (PMID: 37290937), and if the RTN is a primary determinant of breathing problems in Kcnq2 cKO mice (Fig. 4) and Kcnq2 GOF (Fig. 5) during the light/inactive state, then a diminished role of the RTN during wake may contribute to the lack of observed breathing differences between genotypes during the dark/wake state. We have modified the discussion to clarify this point.

4) are the respiratory problems in DEE patients manifest only in the inactive state?

Unfortunately, this is an understudied topic so the answer to this question is yet to be determined. The paucity of literature describing respiratory problems in DEE patients appears to be, at least in part, due to lack of awareness of the potential. Therefore, we hope that a better understanding and greater awareness of the respiratory phenotypes of Kcnq2 GOF and Kcnq2 cKO mice will prompt clinicians to look for a similar phenotype in DEE patients. This point has been added to the discussion.

5) The cell selection process for defining a recorded cell as an RTN neuron could be clearer. It is stated that “Chemosensitive RTN neurons were identified based on location within the ventral parafacial region (Fig. 5A), a minimum 0.5 Hz excitatory response to 10% CO₂, and in some cases PHOX2B immunoreactivity or Phox2b-dependent tdT expression (Fig. 5A)” but then in the

next sentence “However, approximately half of RTN neurons in slices from KCNQ2 GOF mice were not spontaneously active under control conditions and those that were spontaneously active were less active under control conditions (0.7 ± 0.2 Hz, $p = 0.014$) and showed a blunted response to 10% CO₂ compared to RTN neurons in control tissue (1.0 ± 0.1 Hz, $p = 0.0003$) (Figs. 5C-E).” So, at least some of those GOF neurons did not meet the required criterion for RTN identification (delta FR of 0.5 Hz). Then, later, “Neurons that did not show a minimum CO₂ response (even after adjusting baseline activity) were excluded from further analysis” but it is not clear what this minimum CO₂ response threshold is for inclusion.

Sorry for the confusion. To clarify, all RTN neurons included in this study (regardless of genotype) showed at least a 0.5 Hz firing response to 10% CO₂. For those RTN neurons that were not spontaneously active (~50% of neurons from Kcnq2 GOF mice), we delivered a positive DC current to increase baseline firing to an amount similar to control neurons. Then while maintain that current, we tested CO₂ sensitivity. We have clarified this in the text.

6a) Abstract Lines 52-54: “We find that respiratory neurons in the RTN that express the transcription factor PHOX2B also preferentially express KCNQ2 in the absence of other KCNQ isoforms, thus explaining why disruption of KCNQ2 but not other channel isoforms results in breathing problems”. The first part of the statement is on solid grounds but the second one (thus explaining why etc.) seems far-fetched. The authors do indeed show that KCNQ2 mutations alter the properties of RTN neurons in tissue slices in a manner that could potentially account for the changes in whole body chemoreflex but they provide no evidence that RTN neurons are the only Phox2b -dependent neurons that express KCNQ2 and regulate breathing.

Kcnq2 channels are ubiquitously expressed typically in conjunction with Kcnq3 where it functions as a KCNQ2/3 heteromeric channel (PMID: 9836639; PMID: 10684873; PMID 10781098; PMID: 24599470). A striking finding of this study is that RTN neurons break from this trend by expressing Kcnq2 at much higher levels than Kcnq3 (see new data included in Figure 1). Based on this, we speculate that RTN function is uniquely susceptible to Kcnq2 channel dysfunction (no other Kcnq channels to dilute effects of Kcnq2 dysfunction). Conversely, the lack of Kcnq3 transcript in RTN neurons (Fig. 1A) may explain why mutations in this channel have minimal effect on breathing (PMID: 31177578). Further, KCNQ3 is typically expressed with Kcnq2 in other in other respiratory areas including the pre-böttinger complex (PMID: 31824331; PMID: 28819234), hypoglossal motor nucleus (PMID: 29736957), LC (Fig. 1B) and NTS (Fig. 1C), and since mutant Kcnq gain of function channels can heteromerize with wild type subunits to form heteromeric channels (PMID: 35780567; PMID: 31177578; PMID: 25740509), it is reasonable to predict the impact of Kcnq3 mutations expressed in respiratory neurons to be reduced by heteromerization with wild type Kcnq2. We have modified the discussion to clarify this point.

6b) Also, are the authors sure that neurons that express KCNQ2 along with other isoforms are unaffected by these KCNQ2 mutations? The fact that mutations of KCNQ isoforms other than KCNQ2 do not produce the same phenotype does not seem to provide a solid argument against the latter possibility.

We apologize for the confusion. Kcnq2 GOF would also affect neurons that express other isoforms, the extent of which will depend on multiple factors including the presence of other Kcnq channels. However, we expect neurons that only express Kcnq2 would be most affected. Since RTN neurons are the only cell type that fits this profile, we consider the RTN vulnerable to Kcnq2 loss or gain of function mutations. Importantly, since RTN neurons show only modest Kcnq3 expression, we speculate that RTN function is not sensitive to Kcnq3 mutations. Furthermore, since the RTN is an important respiratory center, this expression profile is also consistent with the lack of a respiratory phenotype associated with Kcnq3 mutations. We have modified the discussion to make this more clear.

7a) Lines 84-86: these statements should be less assertive. There is still no direct evidence that the RTN is absent in CCHS patients. This is a hypothesis based on a mouse genetic model (primarily Phox2b27ala and related Phox2b27alacki). Also, the Dubreuil paper (ref 14) actually shows that the pan-Phox2b (Phox2b27ala) mutation is lethal in mice (as it would be in humans without respiration assistance) and the same authors have later shown that a presumably complete and much more selective genetic lesion of the RTN (Phox2b27alacki) does not produce life threatening sleep apnea (since the mice uniformly survive); in fact this latter RTN hypoplasia model exhibits an incomplete (60%) reduction of the hypercapnic chemoreflex in adulthood (Ramanantsoa et al.).

We appreciate the reviewers point. The RTN is a likely contributing factor to breathing problems in CCHS; however, this has yet to be established in CCHS patients and mouse models of this disease suggest that other Phox2b dependent respiratory centers may contribute to baseline breathing problems and early mortality. For example, the noted work by Ramanantsoa et al (PMID: 21900566) showed that restricting the expression of *Phox2b^{27Ala}* to the *Egr2* lineage, which includes the RTN, disrupted RTN development and eliminated the ventilatory response to CO₂ during the neonatal period; however, these animals breathe normally in room air, maintained normal blood gases and survived into adulthood due in part to compensation by peripheral chemoreceptors. Considering that global expression of *Phox2b^{27Ala}* causes lethality at birth, these results suggest that Phox2b-expressing neural populations outside the *Egr2* domain (including the RTN) contribute to features of CCHS.

To clarify, there is little doubt that CCHS is caused by mutations in Phox2b and the Dubreuil et al. (PMID: 18198276) paper noted by the reviewer showed that *Phox2b^{27Ala/+}* mice lack Phox2b expressing glutamatergic neurons in the parafacial region near what is consider the RTN region, whereas other respiratory centers (carotid body, medullary raphe, LC, pre-botC) appeared anatomically normal. Therefore, our statement “since CCHS is caused by variants in the paired-like homeobox 2b (*Phox2b*) transcription factor that is expressed by chemosensitive RTN neurons¹¹⁻¹³, and since the RTN is the only respiratory center perturbed in a mouse model of CCHS (*Phox2b^{27ala/+}*)¹⁴” is not overly assertive.

7b) Finally, in adult rodents, virtually total lesions of the RTN produce hypoventilation but do not produce sleep apnea.

Yes, targeted ablation of ventral parafacial Phox2b+ neurons in rats resulted in hypoventilation but in the absence of disordered breathing during sleep (PMID: 29667182). Consistent with this, we

report here that mice expressing *Kcnq2* R201 in *Phox2b* cells also do not exhibit an apneic phenotype. However, it should be noted that a recent study using the newly developed *Nmb*^{Cre/+} line to target the RTN for caspase-3 lesion showed that ablation of ventral surface *Nmb* cells resulted in baseline (air) hypoventilation during all vigilance states, but particularly NREM sleep, and increased apneic events when breathing high O₂ to limit peripheral chemoreceptor drive (PMID: 37290937). Therefore, contributions of the RTN to baseline breathing remain somewhat unclear.

8) *Results section entitled: “chemosensitive RTN neurons preferentially express Kcnq2 transcript”* The authors show that only 57-66% of RTN neurons (identified as positive for *Slc17a6*, *Phox2b*, *Nmb*, mRNA and the defining proton sensors *Gpr4* and *Kcnk5* mRNA) express *KCNQ2* transcripts. Other *KCNQ* subtypes could not be detected. The *KCNQ3* isoform was generally undetectable. The authors interpret these data as suggesting that *KCNQ2* is selectively expressed by RTN neurons and propose that the channels are homomeric in this cell type, unlike in other neurons. The demonstration is not totally convincing; a quarter of the *KCNQ2*-positive RTN neurons also contained detectable levels of *KCNQ3* mRNA (lines 512). *KCNQ3* mRNA may simply be present at a lower level than *KCNQ2* or may be less effectively detected by hybridization probes. It would be nice to know whether the existence of a high *KCNQ2* / *KCNQ3* mRNA ratio is a unique characteristic of RTN neurons or a feature present in most other neurons. For example, was *KCNQ2* mRNA present at about the same level as *KCNQ3* mRNA in the facial motor neurons or was it much higher as was the case in RTN neurons? In short the evidence that “homomeric *KCNQ2* channels are preferentially express by RTN neurons” (line 266) is not entirely convincing.

Thank you for highlighting this point. As noted above, additional experiments were performed to determine *Kcnq2* and *Kcnq3* transcript expression in several *Phox2b* populations (same probes were used for all regions). We found that the majority of *Phox2b* expressing neurons in the 7N, LC and NTS express both *Kcnq2* and *Kcnq3*. These results have been added to figure 1.

To further address the reviewers concern, we tabulated the number of cells per slice (*Phox2b* reporter mice) from each region of interest that express *Kcnq2* mRNA alone, *Kcnq3* alone, or both *Kcnq2* and *Kcnq3*. We plotted those results as the proportion of cells that only express *Kcnq2* transcript per slice (each dot represent one slice and 100% reflects all cells in that slice express *Kcnq2* in the absence of *Kcnq3*). The figure shows that *Phox2b* expressing neurons in the RTN contain a substantially higher number of *Kcnq2*-only cells per slice, as compared to other populations. These results are included in the text and illustrated as Figure 1F.

9) *Results section entitled: KCNQ2 loss- and gain-of-function variants differentially effect baseline breathing and the central chemoreflex. Lines 145-146, Figure 3 and Supll Figure2.* The authors focus on the respiratory quotient (VCO_2/VO_2) which is basically a measure of which fuel (lipids vs glucose) is preferentially consumed for energy production as opposed to a direct measure of metabolic activity. It seems that reporting the Ve/VCO_2 ratio might have been more informative in the context of this study. Resting VCO_2 and Ve/VCO_2 ratio would help determine whether the mutations cause hyper or hypoventilation at rest, a diagnosis ideally to be confirmed by direct measurements of the arterial PCO_2 . For example, although resting Ve is no different between wildtype and GOF mice (Figs 3 and 4), the GOF mutant mice could very well

hypoventilate at rest (i.e. when no CO₂ is added) meaning that V_e might be the same as that of the controls only because of a rise in steady state PaCO₂. This is typically what happens after moderate size RTN lesions in rodents. Such a result would be consistent with a lower activity of RTN neurons relative to the prevalent PaCO₂ in the GOF mutants (Figure 5B) and would strengthen the authors' notion that the breathing defect originates in part from the presence of mutated KCNQ2 channels in RTN neurons.

We appreciate the reviewer point. However, we are not currently able to measure O₂ consumption (VO₂) or CO₂ production (VCO₂) while simultaneously measuring respiratory activity so we have identified this limitation in the discussion section. To lessen this concern, we expanded our assessment of baseline metabolism to include measures of VCO₂ and VO₂. Consistent with normal baseline breathing, we found that KCNQ2 GOF mice exhibit normal diurnal patterns of metabolic activity. These results have been added to supplemental figure 3.

Minor

10) The text should probably state explicitly that the same light, inactive-state data from GOF neurons in Fig. 3 is re-plotted again with the corresponding dark, active-state data from GOF neurons in Fig. 4.

Thanks for pointing this out. We have clarified this point in the Figure 4 legend.

11) Running title. Please define somewhere what you mean exactly by chemoreception and consider modifying this title or not, accordingly. Regrettably, the word chemoreception is used in the literature in reference to anything from molecules (proton receptors) to whole animal breathing reflex. The present data indicates that the RTN response to CO₂ is sensitive to the channelopathies but is the effect of protons on membrane conductance affected? What changes seems to be cell excitability and the ability to transduce pH stimuli into increased firing. You can define the whole RTN cellular response to CO₂ as chemoreception if you wish but it better be defined.

Thank you for bringing this to our attention. To minimize confusion, we changed the running title from "RTN chemoreception" to "RTN function".

13) The discussion of channel subunit stoichiometry seems overly specified, especially since the relative expression, assembly affinity, etc. of the different variant and isoform subunits is unknown. The authors do qualify that their numbers apply if assuming a binomial distribution, but their point is valid and understandable even without the pseudo-quantification.

We agree and have omitted this discussion.

14) There is no description of how plethysmography experiments were performed at different times of day. In fact, the text says "All experiments were performed between 9 a.m. and 6 p.m. to control for potential circadian effects." So, did someone come in late at night? Were the mice put in rooms with inverted light:dark schedule?

Thanks for catching this oversight. Our standard animal housing conditions use a 12:12 light/dark cycle with lights on at 7a and behavior experiments performed during the light/inactive state were performed between 9a and 5p. For dark cycle experiments, mice were acclimated for at least 1 week in housing with an inverted 12:12 light:dark schedule with lights on at 9p and experiments were performed between 11a and 7p. We have added these details to the methods and updated the results accordingly.

15) What is the Phox2b-Cre line? The text designation as Phox2b^{+/Cre} makes it seem like a knock-in mouse (i.e., Cre at the Phox2b locus) but the mouse line classically used from Jax is a BAC transgenic? Please describe the line and annotate properly if a BAC transgenic.

We used Phox2b-Cre BAC transgenic mice (B6(Cg)-Tg(Phox2b-cre)3Jke/J (JAX stock # 016223). For simplicity, we define these mice as Phox2b::Cre BAC transgenic mice.

16) Fig. 1 legend is not right. Also, the text says that Kcnq3 expression in nearby facial motoneurons is “not shown” but there appear to be some neurons in 7N of Fig. 1B.

We have modified this figure and corresponding legend to include assessment of KCNQ2 and KCNQ3 transcript in other Phox2b neural populations including the facial motor nucleus (7N), LC, and NTS.

17) Fig. 5 legend, “B, traces of firing rate and segments of membrane potential from chemosensitive RTN ...” should be revised since “membrane potential” is not measured in these cell-attached recordings.

It is possible to measure membrane potential in the cell attached configuration (PMID: 16554092). Note that spike amplitude is truncated so we only analyze firing rate in cell attached mode.

18) Since this is the first description of these GOF mice, and other PIs might be interested, please provide supplemental data showing the mis-splicing that is referenced in the text (or a more complete description of how that was determined).

We suspect insertion of stop-flox R201C results haploinsufficiency for two reasons. First, we found that breeding *Kcnq2^{cR201C/+}* mice together resulted in *Kcnq2^{cR201C/R201C}* in only 1 out of 44 mice (6 litters), the one homozygous mouse died shortly after birth. This finding parallels similar patterns seen in the breeding of *Kcnq2* heterozygous null mice, where homozygous mice are born but have a very short lifespan, often succumbing within hours after birth due to failure of breathing (pulmonary atelectasis) (PMID: 10854243; PMID: 12700166). Second, recent work suggests that generation of conditional knock-in mice using the FLEEx switch system may cause deletion of the endogenous exon due to unstable RNA structure (PMID: 30865697). Therefore, we looked for the presence of aberrant splicing in our *Kcnq2^{cR201C/+}* mice. We isolated cDNA from *Kcnq2^{R201C/+}* mouse brains and PCR-amplified it using primers for *Kcnq2* exons 3 to 6. The resulting PCR product was inserted into pGEM-T easy vector and after transformation into NEB® 5-alpha Competent E. coli cells, colonies were screened on LB plates with ampicillin,

IPTG, and X-gal. Selected colonies underwent Sanger sequencing using exon 3 primers, revealing wild-type or exon 4-lacking plasmids, indicating mis-splicing.

To address the reviewers concern, we included a more detailed illustration of the mouse design (Fig. 2A) and we expanded the description of the GOF mice in the methods.

19) p. 12 leading to “higher” resting membrane potential? means more depolarized?

Corrected

20) Please show locations of forward and reverse primers for assessing the GOF mouse. It would also be helpful if the schematic in Fig. 2A also showed the wt allele.

We have added a more detailed schematic of the GOF design that includes primer locations.

21)Supplemental figure 1A/B. Please check the captions (CGG

We have modified the figure captions to be more clear.

Reviewer #2 Comments:

This is an interesting manuscript demonstrating a pathological role of gain of function mutation of KCNQ2(R201C) using conditional knock-in mice that selectively express at RNT neurons. This paper is significant because this determined the cause of breathing problem in patients with KCNQ2(R201C) mutation. Authors demonstrated basic characterization of Phox2b-Cre mediated KCNQ2(R201C) knock-in mice. Followed by functional characterization of respiratory regulation in these knock-in mice. These knock-in mice had normal basal respiration, while had less CO2 induced hyperventilation at light/inactive state. These behavioral differences were confirmed in brain slice experiments. All experiments are logically connected, and results are convincing and they are in good quality. Conclusion and discussion of this paper is valid and supported by the robust experimental results with appropriate analyses.

We thank the reviewer for their time and thoughtful suggestions.

Major comment.

One thing that this reviewer felt missing in this manuscript is that whether this pathological phenotype in KCNQ2(R201C) mice can be treated by ML 252 in whole animal. The authors already use ML252 in their brain slice experiments in Fig. 6. This proposed experiment will elucidate whether KCNQ2(R201C) is the only cause of this phenotype. If they can demonstrate ML 252 can be used to treat GOF mice, this manuscript will have a greater significance on potential therapy.

Great suggestion! It should be noted that although there is some evidence ML252 delivered I.P. can reach the brain (PMID: 22793372), there is a scarcity of published studies describing effects of ML252 on animal behavior. Therefore, to test this treatment we chose to use a high dose of 30mg/kg (dissolved in DMSO and diluted in saline for I.P. injection). Unfortunately, this

treatment had no discernible effect on breathing in control or Kcnq2 GOF mice (N=6 mice per genotype). The lack of any observable change in behavior is surprising and in the absence of a positive control we are not confident ML252 reached the brain. Therefore, we do not consider these results ready for publication.

As an alternative approach, we also attempted to rescue breathing problems in Kcnq2 GOF mice with XE991, a pan Kcnq channel blocker that has been used extensively in vivo (PMID: 26348896; PMID: 32554550). We chose to test a dose of XE991 (2 mg/kg) well below a dose (5 mg/kg) has been shown to cause seizures in wild type mice (PMID: 19921704). At this concentration XE991 (I.P) had minimal effect on breathing in control mice (n=6); however, the first two Kcnq2 GOF mice treated with XE991 developed tonic/colonic seizures and died within minutes of drug injection. Since these animals are in short supply, we did not continue these experiments as it would require us to test multiple XE991 concentrations using multiple cohort of animals to determine optimal dosage. Therefore, at this point, we cannot conclude whether blocking Kcnq channels through systemic injection of a Kcnq blocker will rescue our phenotype.

Specific comments. Some sections of the manuscript were unclear and need revision.

1) *Fig. 1. Legend. Explanation for panel C (B) is missing. Report total cell counts for panel C.*

Sorry for the omission. We have included a detailed description of all figure panels in the updated version. Note that we have modified this figure and corresponding legend to include assessment of Kcnq2 and Kcnq3 transcript expression in multiple Phox2 cell populations including facial motor neurons, LC, an NTS.

2) *Fig. 2C report cell count for each pie chart.*

This information has been added to the figure legend.

3) *Fig. 3 Labeling Bi, Bii, Biii, Ci, Cii, Ciii. I am not sure this labeling system is adequate for journal style. Please check. Also, report n for each genotype.*

We modify the figure labeling to A, B, C, etc. All n's are reported. We also include degrees freedom (n-1) for all statistical comparisons.

4) *Fig. 4 Labeling problems. Same with above.*

corrected

5) *Fig. 5E. This panel does not seem to represent the results shown in panel F. It looks 5HT has less effects in GOF mice.*

There are no genotype differences in 5HT sensitivity. To better illustrate this point we selected a more representative firing rate trace from a control neuron (panel E) and show the summary data as 5HT-change in firing rate.

6) Page 7, line 13-14, & 22-23. Redundant.

We have modified the text to be less redundant.

7) P8. *First sentence. Logic unclear need clarification.*

Kcnq2 GOF mice show normal baseline breathing but a blunted ventilatory response to CO₂ during the light/inactive state. This statement refers to that phenotype being specific to central chemoreception since peripheral chemoreception seems to function normally. To make this clearer we modified the statement to read “Importantly, the respiratory phenotype exhibited by Kcnq2 GOF mice appears specific to a central chemoreception deficit since adult KCNQ2 GOF mice show a normal ventilatory response to hypoxia (10% O₂; balance N₂) (KCNQ2^{+/+} 187 4.1 ± 0.5 ml/g/min vs. Kcnq2 GOF 188 4.0 ± 0.5 ml/g/min; F_{1,14} = 0.11, p = 0.74)”.

8) P9. *ML252, some papers describe as KCNQ2 selective blocker. Why it is describe pan-KCNQ channel blocker in this paper? Need clarification.*

ML252 was originally described as a Kcnq2 blocker (PMID: 22793372); however, subsequent work showed that in fact it blocks several Kcnq isoforms (PMID: 37342413). Therefore, we refer to this drug as a pan Kcnq channel blocker.

9) P11. *Discussion regarding haploinsufficiency is very difficult to follow.*

Sorry for the confusion. We have modified this section of the discussion in the hopes of improving clarity.

10) P11. *Discussion about retigabine and serotonin. Which 5-HT receptors are expressed in RNT neurons? The logic is very difficult to follow. If 5-HT can cause KCNQ2 current suppression, effect of retigabine can be canceled. Related with this, but as mentioned above, it will be very interesting to know whether ML 252 can restore CO₂ regulation in GOF whole animal.*

Serotonin stimulates RTN neurons in part by activation of 5HT₂ and downstream inhibition of Kcnq channels (PMID: 18094252; PMID: 23175845). There is also a modest contribution of 5HT₇ (PMID: 35385139) that appears to activate RTN neurons by activation of HCN channels (PMID: 25429115).

It is not clear whether 5HT can offset or cancel the inhibitory effect of retigabine on neural activity. However, previous work showed in DRG neurons that PIP₂ depletion (which will mimic 5HT/Gq signaling) did not prevent retigabine activation of Kcnq channels near resting membrane potential (PMID: 22155935). Therefore, to the extent retigabine activation of Kcnq mimics Kcnq2 GOF, these results suggest 5HT can overcome the inhibitory effects of Kcnq2 GOF. This possibility is supported by cellular evidence showing that 5HT has similar effects on activity of RTN neurons in slices from control and KCNQ2 GOF mice (Fig. 6E-F).

Reviewer #3 Comments:

In humans, KCNQ2 mutations result in severe developmental and epileptic encephalopathy

(DEE) and breathing disturbances that include profound hypoventilation and apnea. How KCNQ2 mutations and these respiratory phenotypes are linked is not known. Soto-Perez et al. hypothesize that alterations in KCNQ2 expressed in Phox2B+ parafacial respiratory neurons underlie the respiratory dysfunction observed in human patients with KCNQ2 mutations. The authors first use transcriptomics, FISH, and IHC to show that KNCQ2 is the sole KCNQ isoform expressed in a majority of Phox2B neurons. The authors then find that baseline breathing and CO2 ventilatory responses are differentially affected in a state-dependent manner in loss and gain of function KCNQ2 transgenic mouse models where the perturbation is limited to Phox2B-expressing neurons. Finally, the authors use single cell electrophysiology to demonstrate that the GOF KCNQ2 variant alters intrinsic properties of Phox2B neurons in ways that are consistent with the behavioral effects. The authors conclude that alterations in KCNQ2, particularly gain of function mutations, in Phox2B neurons play a role in regulation of breathing that may underlie the respiratory phenotypes seen in humans with KCNQ2 mutations.

The authors extend their previous work on the effects of pharmacological blockade of KCNQ on breathing and address a major gap in our understanding of a clinically relevant channelopathy. The experiments shed light on the specific role of KCNQ2 in the neural control of breathing using multiple approaches that rigorously characterize the effects of a highly specific perturbation from single neuron functional properties to in vivo behavior. The use of both loss and gain of function mouse models that are conditionally expressed in the Phox2B population powerfully exposes state-dependent roles of KCNQ2 in these neurons that provide a link between genotype and phenotype. The in vivo breathing assays and single cell electrophysiology in vitro are robust and the findings are consistent with both their previous work and the clinical presentation in humans. The data are high quality and are clearly presented in text and figures. Analysis and statistics are appropriate; although, a few minor changes (see below) would strengthen the manuscript. The data support their conclusions, and the manuscript will be of interest to basic neuroscientists interested in K+ channels and neural control of breathing and may also be translationally relevant for those studying channelopathies and conditions with respiratory phenotypes.

No major issues noted.

We thank the reviewer for their time and thoughtful suggestions

Minor comments:

1) Statistics. A statistics paragraph in methods would be helpful to understand some of the statistical details throughout the manuscript. Specific issues are listed below.

We apologize for this omission. We have added this necessary paragraph to the methods.

1a) The rationale behind the use of the different post-hoc tests for the various ANOVAs should be described. Tukey, Sidak, and Dunnett's are all reported, and it is not obvious why different post-hoc tests were chosen.

Thanks for requesting this clarification. We have simplified our analysis of ANOVA results to just the Tukey's multiple comparison test.

1b) A statement about how the data were determined to be normally distributed should be provided to justify the exclusive use of parametric tests.

Normality was determined using the Anderson-Darlington, D'Agostino & Pearson, and Shapiro-Wilk tests. Data were considered non-normal if it failed all three normality tests and analyzed using the Mann-Whitney test followed by the Holm-Sidak multiple comparison test. The non-parametric analysis of covariance was used to compare slopes of linear regressions. The specific test used for each comparison is reported in the figure legend and all relevant values used for statistical analysis are included in the results section.

1c) Both F statistic and p-value should be reported wherever an ANOVA is run.

All F (T and H) and p values have been added to the results section.

1d) Details of the group data figures, bar, scatter, and box plots, particularly whisker values and what error bars represent (sd or sem), should be reported.

Sorry for this confusion. We have included n's for all summary data and all data reported in the results section are as mean +/- SEM. All summary data are plotted as mean (not median) with individual data points. The box plot whiskers show maximum and minimum distribution of data, whereas all other error bars show SEM. We have added this to the statistical paragraph in the methods and clearly define error bars in each figure.

1e) What is the rationale behind using box plots for some graphs and mean +/- sd scatter plots for other panels given that all data are, I assume, normally distributed?

Binary/tabulated data are shown as pie charts to best represent proportions of the Phox2b population that express each transcript of interest.

In our opinion box-whisker plots do a better job reflecting the range of data under one condition so most summary data was shown as modified box-whisker plots where the middle line is the mean (not median) and the whiskers show the maximum and minimum range of data.

To minimize business, we chose to depict experiments involving repeated measures over a range of conditions as line graphs with each data point reflecting the mean +/- SEM.

1f) A 2 way repeated measures ANOVA is reported for Fig 6B experiments, but I do not think this test is appropriate for the study design, which involves two independent groups that do not undergo all "treatments."

Thanks for the suggestion. To address this concern, we decided to analyze these results by comparing the area under the curve (AUC) for each condition (control and Kcnq2 GOF before and in the presence of ML252). The AUC provides a measure of the total effect of Kcnq2 GOF

on RTN neuron excitability, which is the parameter we are most interested in. We have reanalyzed these results and modified the text accordingly.

2) p. 5, line 117. *Levels of other Kcnq isoforms in transcriptomic analysis are stated “not shown”, but Fig 1A appears to show the data and should be referenced. Figure 1 legend also omits a description of Figure 1A.*

We have modified Figure 1 to focus on fluorescent in situ analysis of Kcnq transcripts in multiple Phox2b populations. The qPCR results are mentioned in the text but are no longer illustrated.

3) p. 6, lines 142-150. *Authors note the lower than expected frequencies of mutants and conclude that both cKO and GOF may be associated with embryonic lethality. The expected Mendelian frequencies for both genotypes (50% for cKO and 25% for GOF?) and number of litters collected for this analysis should be reported. The possible causes of embryonic lethality should be discussed given the relatively mild phenotype of the animals that survived and the fact that breathing is not necessary for survival in utero. Is there any evidence for perinatal, rather than embryonic, lethality?*

Thank you for bringing this to our attention. The proportion of mice born alive was determined within 24 hours of birth so we cannot differentiate embryonic lethality from death within hours after birth. We have modified the text to read “Although the proportion of KCNQ2 GOF (15% of 31 litters) and KCNQ2 cKO (18% of 19 litters) mice in each litter was less than expected (determined within 24 hr of birth), postnatally...”.

4) p. 6, line 152-3. *Based on this sentence, it seems like the analysis of baseline breathing in cKO is limited to the light/inactive state, yet the specificity of the difference in baseline breathing to the light/inactive phase (or that it was only measured during light) is not apparent when this result is presented in abstract, introduction, Figure 3, and discussion. If analysis was restricted to light/inactive, this should be made explicit throughout. Given the increase in serotonin during wakefulness and its effects on KCNQ, it would be interesting to note the effects, if any, of the cKO on baseline breathing and CO₂ response during the dark/awake state and on sighs and apneas for both states.*

As suggested, we performed additional experiments to characterize Kcnq2 cKO breathing during the dark/active state. We found that Kcnq2 cKO mice which have high baseline breathing in the light/inactive state show normal baseline breathing during the dark/active state (Fig. 4D). These results are similar to the light/inactive-state specific chemoreceptor deficit exhibited by Kcnq2 GOF mice (Figs. 5E-F) and together suggest that inhibition of Kcnq2 by wake-on neuromodulators limits breathing problems associated with loss or gain of Kcnq2 channel mutations. These results are shown as a new figure 4 and the results and discussion have been modified accordingly.

5) p. 8, line 199-203. *Is the firing rate for 10% CO₂, the change in firing rate? The values as reported are very similar and figure 5D shows a delta CO₂. This should be clarified in the text and figure legend.*

We confirm that the text and figure refer to CO₂ induced changes in activity. We have modified the text to make this clearer.

6) p. 11, line 266. On p. 12, line 289-294, the role of serotonin in perhaps limiting the GOF effect to light state is discussed. A complementary discussion on serotonin and cKO would be similarly enlightening. From the Hawryluk et al. 2012 paper, XE991 had an effect on baseline breathing in anesthetized, but not awake rats, though KNCQ blockade blunted 5HT responses in both states. It seems that discussing how a blunted serotonin effect in cKO might be compensated for in awake states would be interesting.

Thanks for this suggestion. We have modified this section of the discussion as follows “Our evidence that Kcnq2 loss or gain of function in Phox2b-expressing neurons disrupted breathing during the light/inactive but not dark/active state is consistent with previous work showing that injection of XE991 (pan Kcnq channel blocker) into the RTN of anesthetized rats increased baseline breathing, where this same manipulation in conscious rats had negligible effect on breathing¹⁹. The basis for potential circadian differences in behavioral responses to RTN specific Kcnq channel manipulations are not clear. One possible explanation is that the RTN has a dominate role in control of breathing during times of reduced arousal, whereas during wakefulness contributions of the RTN to breathing may be diluted by numerous other excitatory inputs to the system. Consistent with this, targeted ablation of RTN neurons using a newly developed Nmb^{Cre/+} mouse resulted hypoventilation under all vigilance states, but most profoundly during NREM sleep³⁷. Another possibility is that inhibition of Kcnq2 channels by wake-on neuromodulators limits the impact of mutant channels on RTN function and breathing. For example, inhibition of Kcnq2 by wake-on neuromodulators may diminish genotype differences in channel function and so lessen the impact of loss of Kcnq2 function on baseline breathing in the dark/active state (Fig. 4D). Also, since RTN neurons that express Kcnq2^{R201C} respond normally to serotonin (Figs. 6E-F), we speculate that inhibition of Kcnq2^{R201C} by wake on neuromodulators offsets the gain of function effect on respiratory output in the dark/active state. Conversely, when arousal signaling is at its nadir during the light/inactive state, RTN neurons are subject to the full gain-of-function effect of Kcnq2^{R201C} which manifests as a diminished ventilatory response to CO₂ (Fig. 5E). However, it remains to be determined whether breathing problems in DEE patients manifest in a sleep-wake or circadian dependent manner”.

7) p. 15, lines 374-375. The authors note that all experiments were performed between 9 am and 6 pm and earlier note that animals were housed with a 12:12 light/dark cycle. The time of the switch should be reported to make this statement meaningful and illuminate when the dark/active experiments were performed.

This information has been added to the methods.

8) p. 21, line 515. Figure 2 legend title is unclear.

Thanks. It now reads “General characterization of Kcnq2 cKO and Kcnq2 GOF mouse models”

9) p. 22, line 537. “effects” should be “affects”
fixed

10) *Figure 1A. Using a bar graph with n=3 data points that are almost all overlapping is a bit unclear. It might be clearer to just plot the individual data points and a scatter plot with mean +/- sd.*

Thanks for the suggestion. We chose to focus Fig 1 on fluorescent in situ expression of KCNQ2 and KCNQ3 transcripts in various Phox2b neural populations. The qPCR results are not included in the text in lieu of an illustration.

11) *Figure 2c. Scale bar value missing.*

thanks. This value has been added to the figure legend.

12) *Figure 4Bi, legend, "Kcnq2" should be capitalized.*

Fixed.

13) *Figure 4Ci, ii, "bal.O2" in Ci and "balance O2" in Cii should be made consistent.*

Done.

14) *Figure 5A right, scale bar value missing.*

That is defined in the figure legend.

REVIEWER COMMENTS

Reviewer #1 (Remarks to the Author):

This study offers interesting and generally well-documented facts regarding the effects of KCNQ2 mutations on the intrinsic properties of RTN neurons and on their response to CO₂ in slices. However, the authors attempt to attribute the breathing disturbances exhibited in vivo by these mutant mice selectively to cell autonomous changes in the properties of the RTN neurons is unconvincing; this interpretation should be presented as a mere hypothesis to be tested in the future.

MAJOR

- 1) The authors' three lines of evidence presented to shore up their theory (and summarized below) must be discussed more objectively and the conclusion must definitely be toned down.
- 2) First evidence: the intrinsic properties of RTN, including their response to CO₂, are affected by the KCNQ2 mutations. This is correct and constitutes the non-controversial aspect of the present report. However, this evidence does not prove that the changes in RTN properties observed in neonatal brain slices account, even partially for the breathing disturbances observed in the intact animals. This point could only be made by introducing these mutations solely to the RTN neurons. This issue was mentioned in the original review but was not addressed in revision.
- 3) Second point: the disease CCHS can be explained by the sole malfunction of the RTN based on mouse genetic studies, therefore, the breathing disturbances of mice with mutated KCNQ2 are likely caused by changes in the properties of RTN neurons. This rationale rests on an outdated theory according to which *Phox2b* mutation (*Phox2b27ala/+*) prevents the development of the RTN without causing any other deleterious effect on the breathing network of mice (details later).
- 4) Third point: RTN are the only neurons that express KCNQ2 but not KCNQ3 whereas all other neurons express both, therefore the defects associated with KCNQ2 mutations must be attributed to these neurons. This is hardly convincing both theoretically and based on the data presented. Firstly, thousands of neurons compose the respiratory pattern generator and contribute neuronal input to it. How can one exclude the potential existence of myriad neurons that also express selectively KCNQ2 and influence the chemoreflex or breathing patterns? The gain of the chemoreflex depends on the reactivity of the entire breathing network downstream of the RTN neurons and, like RTN neurons, this network is modulated by innumerable inputs, notably serotonin, orexin, noradrenergic and other inputs which contribute to the chemoreflex in some fashion. For example, the data presented in Fig 1A, show that facial motoneurons express *Phox2b*, high levels of KCNQ2, and much less KCNQ3, the same pattern described for RTN. Furthermore, the authors' histological evidence of lower level of expression of KCNQ3 than KCNQ2 in RTN is modest at best given that even the KCNQ2 signal is barely detectable (Figure 1A, RTN; more later). Finally, while the authors provide excellent evidence of the presence of KCNQ2 protein (by IHC with evidence that the signal is absent in the KO mouse), the presence or absence of KCNQ3 protein is not documented by the same means. Finally on this point, the paper discusses possible heteromerization of KCNQ2 with KCNQ5 subunits, but there are no in situ data presented on expression of those subunits in *Phox2b*-expressing neuron populations. Is KCNQ5 expressed in RTN? If so, would the authors logic suggest that this would dilute the effect of the KCNQ2 mutations, as presently argued for the other KCNQ3-expressing *Phox2b* neurons? This was apparently tested by qPCR from cell sorted material, but those results may have been contaminated by other cells (see below). The authors should examine expression of KCNQ5 by RTN neurons using RNAscope.
- 5) "loss and gain of *Kcnq2* function in *Phox2b* expressing RTN neurons preferentially disrupted baseline breathing or the central chemoreflex, respectively, during the light/inactive state". This conclusion seems more "aspirational" than justified by the data; causality between the changes in RTN properties and the pathological breathing defects associated with these mutations is not supported by hard evidence. Besides, there is increasing evidence that epilepsy affects breathing via descending inputs originating in the forebrain (<https://pubmed.ncbi.nlm.nih.gov/32163374/> among others).

Other issues (in no particular order).

1. Please settle on a definition of the RTN at the outset. The neurons are identified as "parafacial respiratory neurons" (title), RTN chemoreceptors (is there another type of RTN neuron?), RTN

neurons, RTN Phox2b neurons (are there Phox2b-negative RTN neurons? Are there acid-insensitive RTN neurons? How about defining RTN as a cell type? Neurons positive for Phox2b, NMB, Vglut2, TASK-2, GPR4, negative for a slew of marker (TH, TPH, GAD GlyT2 etc...) and with a defined developmental lineage (*egr-2/Phox2b/Lbx1/Atoh-1*)?

2. First part of the results and Figure 1. The authors identified Phox2b-expressing cells by the presence of TdT in a Cre-BAC x TdT-reporter mouse cross. With this approach TdT tags neurons and other cells (e.g., astrocytes? PMID: 30707772) that have expressed Phox2b at any time during development; a lot of them no longer express this transcription factor and this could be the case of some or all of the neurons shown "below the facial motor nucleus" and identified as RTN. This could also be the case for the cells (neurons?) obtained by cell sorting.

3. Figure 1A and specifically RTN lower left panel must be revised. The KCNQ2 signal is barely detectable in the cells (neurons?) identified by the white arrows in the "merged panel" even when using 500% magnification of the PDF file. Bottomline: Figure 1A does not illustrate the authors' point which is that RTN neurons selectively express KCNQ2. The ISI signals are so low that concluding on that basis that the cells in the marginal layer selectively express KCNQ2 stretches credibility. If this is a technical issue with the figures, it must be sorted out by using higher magnification and by showing that the KCNQ2 signal co-localize with neurons that actually contain Phox2b immunoreactivity or Phox2b transcripts. Also, in Fig. 2 the loss of KCNQ2 immunoreactivity in the cKO is credible but those Phox2b+ cells don't look much like RTN neurons. They could be facial motoneurons or other cells. The phenotype of these putative Phox2b+ cells should be confirmed using one of several known diagnostic markers of the RTN neurons.

4. Line 127: Phox1B+ facial motoneurons?

5. Line 89-90. "since the RTN is the only respiratory center perturbed in a mouse model of CCHS (*Phox2b27ala/+*)¹⁶". This rationale is not supported by more recent work from the same group. The quote is lifted from a seminal 2008 paper (ref 16) in which the authors first showed that RTN neurons fail to develop in a *Phox2b27ala/+* mouse, a CCHS model; it is true that the authors were not able to identify any other morphological changes besides the lack of RTN neurons but this does not mean that there were no other defects elsewhere. In their later work (Ramanantsoa et al. J. Neuro 2011, not cited here), the same group showed that expressing *Phox2b27ala* selectively into cells of *egr-2* dependent lineage also eliminates RTN neuron development. However, all these mice survive to adulthood whereas the *Phox2b27ala/+* mice (of ref 16) died within hours of birth. This evidence shows convincingly that the RTN is very unlikely to be the only respiratory "center" perturbed in the *Phox2b27ala/+* mouse model of CCHS and that this mutation impacts additional brain neurons that are essential for breathing and survival.

6. Lines 160-168: how did the authors assess the effect of CO₂ during the dark /active state? What was the baseline? Typically, the breathing stimulation elicited by CO₂, and possibly caused by RTN activation, can only be measured with any reliability during periods when breathing is regular (animals are resting) as opposed to ambulating, sniffing, eating etc...

7. Line 212: "and in some cases Phox2b immunoreactivity or Phox2b-dependent tdT expression". As already mentioned, the presence of TdT does not identify RTN neurons specifically; TdT could be present in any neuron that has expressed Phox2b at any point during development (e.g., facial motoneurons, astrocytes). Also, in neonate mice, Phox2b immunoreactivity is still detectable in facial motor neurons. In any event, the authors should report what proportion of presumptive RTN neurons were actually identified using these histological criteria.

8. Line 214-215; many would argue that an increase in firing rate of 1.7 Hz from a baseline of 1.8Hz when CO₂ is increased from 5 to 10%, is anything but "robust" even though it may be typical of RTN neurons in the authors' experimental conditions,. Similar or larger increases have been reported in many other neurons (NTS, serotonin, Locus coeruleus). Does a more alkaline reference point (3% CO₂?) increase the dynamic range of the response to acidification?

9. Lines 229-230: "These results show that expression of *Kcnq2R201C* in *Phox2b+* neurons caused cell autonomous suppression of baseline firing and CO₂/H⁺ stimulated activity". Why is the effect of this mutation on RTN neurons deemed "cell-autonomous"? This could be true of course but no attempt was made at isolating RTN neuron synaptically. Besides, this and other labs have previously published evidence that, in slices, RTN neurons are at least partly regulated via CO₂-sensitive synaptic inputs

(serotonergic, SST-ergic). <https://pubmed.ncbi.nlm.nih.gov/30866045/>;
<https://pubmed.ncbi.nlm.nih.gov/34013884/>.

10. Lines 481-482: This should likely state that the anesthetized animals were perfused with paraformaldehyde before the brains were removed? or was the RNAscope and IHC performed on unperfused tissue?

Reviewer #2 (Remarks to the Author):

Authors responded to all my concerns raised from the previous edition. The experiments that I suggested turned inconclusive, but it is understandable that it could be failed due to non-selective nature of pharmacological experiments and compound-specific limitation. However, readers deserve to know the finding of ML252 and XE991 in whole animal responses with a discussion of potential causes for negative results. Short statement would be sufficient for this propose.

On the other hand, revised Fig. 1 and Fig. 2 brought new confusions that need to be addressed.

Fig. 1. First of all, does the legend explain correctly regarding white and red arrows? The legend indicates "white arrows identify cells that express KCNQ2 but not KCNQ3 and red arrows identify cells that co-express both transcripts" Beside typo (identity), I could not confirm KCNQ2 transcript in RTN cells with white arrows, but only in red arrows. In addition, it is hard to tell KCNQ3 shows any transcripts in all panels. This reviewer thinks authors need to define the criteria for "positive transcript" in the method section. Also, corresponding arrows should be indicated in KCNQ2 and KCNQ3 panels to show where we should look at. Finally, there is a purple arrow in panel B merge. What is it? For panel D, total cell counts were reported in Supplementary data set as I requested. However, supplementary dataset has never been mentioned in the main text and should be presented in the relevant section. In panel E, reported number and what are shown did not match. For instance, for 7N, I can count 14 symbols but reported n was 12.

Fig. 2 panel B. In my initial review, I thought this was genomic PCR, but the legend reads "transcript", thus suggesting RT-PCR. Is this true? Since Fig. 2A and 2B are barely explained in the main text, it is hard to confirm. This should be clarified. In addition, the revised figure 2B indicates KCNQ2f/f gives shorter band than KCNQ2+/, which contradicts to the previous edition. Which is correct?

Page 8 line 2. After reading comments from reviewer 1, I wonder "diminished RTN chemoreception" should be rephrased to "diminished RTN sensation of CO₂" would be more specific for this paper. Please consider.

Reviewer #3 (Remarks to the Author):

The authors have satisfactorily addressed the major and minor issues in their revision. Missing methodological details are now provided, statistical methods are adequately described, and figures and figure legends are now clearer. The discussion touches on relevant points, and the authors' conclusions are supported by the data presented.

Reviewer #1 Comments:

This study offers interesting and generally well-documented facts regarding the effects of KCNQ2 mutations on the intrinsic properties of RTN neurons and on their response to CO₂ in slices. However, the authors attempt to attribute the breathing disturbances exhibited in vivo by these mutant mice selectively to cell autonomous changes in the properties of the RTN neurons is unconvincing; this interpretation should be presented as a mere hypothesis to be tested in the future.

We thank this reviewer for their support and many helpful suggestions. In short, we agree with all your points. Please see our point-by-point responses below for our detailed responses.

MAJOR

1) The authors' three lines of evidence presented to shore up their theory (and summarized below) must be discussed more objectively and the conclusion must definitely be toned down.

We agree and we thank the reviewer for helping adjust our interpretations to be more objective. We have modified how the data is presented and interpreted to be more specific to the targeted cell type. In particular, we emphasize that Phox2b is expressed by several respiratory centers including but not limited to RTN neurons. We discuss results related to Figs 1-5 in the context of Phox2b-expressing cells (we also changed the title to reflect this focus). Our slice patch experiments targeted cells with properties consistent with RTN neurons so those results were used to better understand how loss and gain of Kcnq2 function impacts that specific type of Phox2b neuron. However, we no longer attribute the respiratory phenotype of Kcnq2 cKO and Kcnq2 GOF only to RTN neurons. Specifically, our main conclusion is that Phox2b-expressing neurons including those in the ventral parafacial region (putative RTN neurons) might be vulnerable to Kcnq2 channelopathies and may contribute to breathing problems associated with KCNQ2 developmental and epileptic encephalopathy. We hope the reviewer agrees that the revised version provides a balanced presentation and interpretation of the data.

2) First evidence: the intrinsic properties of RTN, including their response to CO₂, are affected by the KCNQ2 mutations. This is correct and constitutes the non-controversial aspect of the present report. However, this evidence does not prove that the changes in RTN properties observed in neonatal brain slices account, even partially for the breathing disturbances observed in the intact animals. This point could only be made by introducing these mutations solely to the RTN neurons. This issue was mentioned in the original review but was not addressed in revision.

We apologize if we neglected to address this concern in the previous revision. We agree with the reviewer that targeting all Phox2b+ neurons is not specific to RTN neurons and so more targeted approaches are required to link RTN neural dysfunction to the in vivo phenotype. Unfortunately, for reasons beyond our control, we will not be able to obtain the number of animals necessary for this experiment in a timely manner and so we are unable to perform this experiment at this time. Therefore, as suggested we modified how the data is presented and interpreted with a more objective focus on Phox2b-expressing neurons rather than RTN neurons.

3) Second point: the disease CCHS can be explained by the sole malfunction of the RTN based on mouse genetic studies, therefore, the breathing disturbances of mice with mutated KCNQ2 are likely caused by changes in the properties of RTN neurons. This rationale rests on an outdated

theory according to which Phox2b mutation (Phox2b27ala/+) prevents the development of the RTN without causing any other deleterious effect on the breathing network of mice (details later).

There is little doubt that the RTN is an important contributing factor to breathing problems in CCHS, and since people that express the KCNQ2 gain of function variant R201C exhibit a hypoventilatory phenotype similar to CCHS, we think our focus on the RTN is justified. We agree with your point that other Phox2b-expressing populations likely contribute features of this disease. Therefore, since we are not able to specifically manipulate KCNQ2 function only in RTN neurons (noted in concern #2), we have modified our conclusions to focus on Phox2b-expressing cells and to be less RTN centric.

4a) Third point: RTN are the only neurons that express KCNQ2 but not KCNQ3 whereas all other neurons express both, therefore the defects associated with KCNQ2 mutations must be attributed to these neurons. This is hardly convincing both theoretically and based on the data presented. Firstly, thousands of neurons compose the respiratory pattern generator and contribute neuronal input to it. How can one exclude the potential existence of myriad neurons that also express selectively KCNQ2 and influence the chemoreflex or breathing patterns? The gain of the chemoreflex depends on the reactivity of the entire breathing network downstream of the RTN neurons and, like RTN neurons, this network is modulated by innumerable inputs, notably serotonin, orexin, noradrenergic and other inputs which contribute to the chemoreflex in some fashion. For example, the data presented in Fig 1A, show that facial motoneurons express Phox2b, high levels of KCNQ2, and much less KCNQ3, the same pattern described for RTN.

Facial motor neurons do not show the same pattern of Kcnq expression and RTN neurons; we hope this will be more evident in the updated Fig. 1 which shows facial motoneurons express both Kcnq 2 and Kcnq 3 (red arrows). To our knowledge no other neural population has been shown to selectively expresses Kcnq2. Therefore, expression of Kcnq2 in the absence of other Kcnq channel isoforms is unusual, and since wild type Kcnq3 can heteromerize with Kcnq2 R201C to diminish the gain-of-function effect, it is reasonable to speculate that in the absence of Kcnq binding partners, the RTN will be vulnerable to Kcnq2 channelopathies.

Our manipulations are specific to Phox2b-expressing cells; therefore, we conclude the respiratory phenotype of Kcnq2 cKO and Kcnq2 GOF mice involves Phox2b-expressing cells.

4b) Furthermore, the authors' histological evidence of lower level of expression of KCNQ3 than KCNQ2 in RTN is modest at best given that even the KCNQ2 signal is barely detectable (Figure 1A, RTN; more later).

We appreciate the reviewers point that the intensity of fluorescent *in situ* labeling for Kcnq2 in the RTN appears less robust compared to other brainstem regions investigated. Therefore, to bolster our conclusion that Kcnq2 but not Kcnq3 is preferentially expressed by Phox2b-expressing ventral parafacial neurons, we performed two additional experiments. As described in the first section of results, we re-analyzed a previously published transcriptomic dataset of ventral parafacial neurons (accession ID GSE174417) (PMID: 34645823). For this analysis, RTN neurons were identified based on expression of *Slc17a6*, *Phox2b*, *Nmb* and absence of *Th*; we found that cells with this molecular profile express Kcnq2 transcript but show low or not detectable expression of other Kcnq subtypes (see Review Figure 1). To further validate this finding, we performed qPCR for all five Kcnq isoforms in an enriched population of Phox2b expressing neurons obtained from the ventral parafacial region. Of the five channel isoforms, only Kcnq2 transcript was detectable in this experiment. All three lines of evidence consistently show that Phox2b-expressing neurons in the ventral parafacial region preferentially express Kcnq2. Therefore, we believe this is a reasonable conclusion.

4c) Finally, while the authors provide excellent evidence of the presence of KCNQ2 protein (by IHC with evidence that the signal is absent in the KO mouse), the presence or absence of KCNQ3 protein is not documented by the same means.

To our knowledge, this is the first validation of this commercial Kcnq2 antibody for immunocytochemistry. We attempted to parallel this experiment using Kcnq3 antibodies but unfortunately, we found that all commercially available Kcnq3 antibodies show poor specificity on tissue from Kcnq3 knockout mice. Therefore, unfortunately the suggested experiment is not possible at this time.

4d) Finally on this point, the paper discusses possible heteromerization of KCNQ2 with KCNQ5 subunits, but there are no *in situ* data presented on expression of those subunits in Phox2b-expressing neuron populations.

Is KCNQ5 expressed in RTN? If so, would the authors logic suggest that this would dilute the effect of the KCNQ2 mutations, as presently argued for the other KCNQ3-expressing Phox2b neurons? This was apparently tested by qPCR from cell sorted material, but those results may have been contaminated by other cells (see below). The authors should examine expression of KCNQ5 by RTN neurons using RNAscope.

We performed the suggested experiment and those results are discussed in the text and shown in a new supplemental figure 4. In short, we found very little fluorescent *in situ* labeling for Kcnq5 in Phox2b ventral parafacial neurons. As a positive control, we used the same probe to confirm

expression of Kcnq5 mRNA in the CA3 region of the hippocampus (reviewer Fig. 2). These results are consistent with our qPCR results shown in supplemental Fig. 1E. Based on this, we conclude Phox2b-expressing neurons in the ventral parafacial region express low levels of Kcnq5 transcript. Regarding potential contamination of our sorted cells. We found no evidence for glia contamination of our sorted material (see response to Other Issues #2). It is also worth noting that glial cells have not been shown to express KCNQ5. Also, from a theoretical perspective, if some other cell type did contaminate our sample, the expected outcome would be a false positive rather than a failure to detect Kcnq5 as reported here. Therefore, we do not consider this an issue.

5a) “loss and gain of Kcnq2 function in Phox2b expressing RTN neurons preferentially disrupted baseline breathing or the central chemoreflex, respectively, during the light/inactive state”. This conclusion seems more “aspirational” than justified by the data; causality between the changes in RTN properties and the pathological breathing defects associated with these mutations is not supported by hard evidence.

As noted above, we have modified all conclusions to be less RTN centric. The statement in question now reads “loss and gain of Kcnq2 function in Phox2b expressing neurons preferentially disrupted baseline breathing or the central chemoreflex, respectively, during the light/inactive state”.

5b) Besides, there is increasing evidence that epilepsy affects breathing via descending inputs originating in the forebrain (<https://pubmed.ncbi.nlm.nih.gov/32163374/> among others).

Loss of Kcnq2 in the cortex is associated with seizure activity (PMID: 33768249), cortical seizures may propagate to brainstem to disrupt control of breathing (PMID: 36605420), and preliminary evidence noted in the last rebuttal suggests Kcnq2 GOF mice are prone to seizure induced death in response to systemic injection of a pan-Kcnq channel blocker (XE991). However, Kcnq2 cKO and Kcnq2 GOF mice do not exhibit overt seizures under all experimental conditions tested here. Therefore, it's unlikely cortical seizure actively contributes to the phenotypes described here.

Other issues (in no particular order).

1. Please settle on a definition of the RTN at the outset. The neurons are identified as “parafacial respiratory neurons” (title), RTN chemoreceptors (is there another type of RTN neuron?), RTN neurons, RTN Phox2b neurons (are there Phox2b-negative RTN neurons? Are there acid-insensitive RTN neurons? How about defining RTN as a cell type? Neurons positive for Phox2b, NMB, Vglut2, TASK-2, GPR4, negative for a slew of marker (TH, TPH, GAD GlyT2 etc...) and with a defined developmental lineage (egr-2/Phox2b/Lbx1/Atoh-1)?

We like this suggestion and so used the term ‘RTN neurons’ throughout the text.

We do not always have access to multiple molecular markers to define the cell type of interest so we explicitly state how we define RTN neurons in the two experiments specific to this population. As suggested by the reviewer, we use the presence and absence of various markers to define RTN neurons in transcriptome data. However, for slice electrophysiological experiments we rely on location ventral surface, CO2/H+ sensitivity and when possible Phox2b

expression. As recognized by the reviewer, RTN neurons are the only identified CO₂/H⁺ sensitive cell in the ventral parafacial region so we consider this a reasonable approach.

2. First part of the results and Figure 1. The authors identified Phox2b-expressing cells by the presence of TdT in a Cre-BAC x TdT-reporter mouse cross. With this approach TdT tags neurons and other cells (e.g., astrocytes? PMID: 30707772) that have expressed Phox2b at any time during development; a lot of them no longer express this transcription factor and this could be the case of some or all of the neurons shown “below the facial motor nucleus” and identified as RTN. This could also be the case for the cells (neurons?) obtained by cell sorting.

To control for potential glial contamination of our sorted material we included probes for a neuron specific marker (*Rbfox3*), two astrocyte markers (*Gfap* and *Aldh1l1*) and an oligodendrocyte marker (*Olig1*) in all qPCR reactions. As expected, our sorted *Phox2b* cells showed high levels of *Rbfox3* (NeuN; mean raw CT value 21.97 ± 0.14 ; n=3) and not detectable levels of levels of *Gfap*, *Aldh1l1* or *Olig1*. These results have been noted in the text. We also updated the methods to include the probe details.

3a. Figure 1A and specifically RTN lower left panel must be revised. The KCNQ2 signal is barely detectable in the cells (neurons?) identified by the white arrows in the “merged panel” even when using 500% magnification of the PDF file. Bottomline: Figure 1A does not illustrate the authors’ point which is that RTN neurons selectively express KCNQ2. The ISI signals are so low that concluding on that basis that the cells in the marginal layer selectively express KCNQ2 stretches credibility. If this is a technical issue with the figures, it must be sorted out by using higher magnification and by showing that the KCNQ2 signal co-localize with neurons that actually contain Phox2b immunoreactivity or Phox2b transcripts.

We apologize for the poor quality of the previous Fig. 1 images. We re-imaged at a higher resolution to maximize pixel density; the new ventral parafacial images show two tdT-labeled *Phox2b*-expressing cells that meet our criteria for *Kcnq2* transcript expression (soma that contained 5 or more labeled puncta was considered positive for that transcript). We also confirmed by qPCR (Supplemental Figure 1E) and re-analysis of previously published transcriptomic data (Reviewer Fig. 1) that *Phox2b*-expressing cells in the ventral parafacial region preferentially express *Kcnq2*. These results are further bolstered at the protein level by our evidence that *Phox2b*-immunoreactive cells in the ventral parafacial region from control mice show robust *Kcnq2* immunoreactivity (Fig. 2C). We hope the reviewer agrees that based on these multiple lines of evidence, it is reasonable to conclude *Phox2b* expressing ventral parafacial neurons express *Kcnq2* channels.

3b) Also, in Fig. 2 the loss of KCNQ2 immunoreactivity in the cKO is credible but those Phox2b+ cells don't look much like RTN neurons. They could be facial motoneurons or other cells. The phenotype of these putative Phox2b+ cells should be confirmed using one of several known diagnostic markers of the RTN neurons.

Thank you for bringing this to our attention. We have selected new representative images to include *Phox2b* cells that look similar to RTN neurons (new Fig. 2C). However, in this revised version of the ms we attribute phenotypes associated with *KCNQ2* cKO and *KCNQ2* GOF mice

to Phox2b-expressing cells in general and not specific to the RTN; therefore, we do not consider it necessary to differentiate RTN neurons from other Phox2b-expressing cells in the region of interest.

4. Line 127: *Phox1B+ facial motoneurons?*

Thanks. Fixed.

5. Line 89-90. *“since the RTN is the only respiratory center perturbed in a mouse model of CCHS (Phox2b27ala/+)¹⁶”. This rationale is not supported by more recent work from the same group. The quote is lifted from a seminal 2008 paper (ref 16) in which the authors first showed that RTN neurons fail to develop in a Phox2b27ala/+ mouse, a CCHS model; it is true that the authors were not able to identify any other morphological changes besides the lack of RTN neurons but this does not mean that there were no other defects elsewhere. In their later work (Ramanantsoa et al. J. Neuro 2011, not cited here), the same group showed that expressing Phox2b27ala selectively into cells of egr-2 dependent lineage also eliminates RTN neuron development. However, all these mice survive to adulthood whereas the Phox2b27ala/+ mice (of ref 16) died within hours of birth. This evidence shows convincingly that the RTN is very unlikely to be the only respiratory “center” perturbed in the Phox2b27ala/+ mouse model of CCHS and that this mutation impacts additional brain neurons that are essential for breathing and survival.*

I think we agree that the RTN likely contributes to breathing problems in CCHS. A hallmark feature of CCHS is the lack of a central chemoreflex and the noted study by Ramanantsoa et al. (PMID: 21900566) showed that expression of Phox2b27ala in the Egr2 domain (which includes the RTN but not other Phox2b expressing respiratory regions like the NTS and LC) eliminated the neonatal ventilatory response to CO₂, thus recapitulating a defining feature of CCHS. However, as noted, these mice survive to adulthood, therefore, we also agree that other Phox2b-expressing populations are likely to contribute to aspects of CCHS.

We modified our statement to read “Since CCHS is caused by variants in the paired-like homeobox 2b (*Phox2b*) transcription factor that is expressed by several respiratory centers including chemosensitive RTN neurons¹³⁻¹⁵, and since disruption of the RTN may contribute to diminished respiratory chemoreception in a mouse model of CCHS (PMID: 21900566), we consider the RTN a potential substrate responsible for KCNQ2 encephalopathy related breathing abnormalities”.

6. Lines 160-168: *how did the authors assess the effect of CO₂ during the dark /active state? What was the baseline? Typically, the breathing stimulation elicited by CO₂, and possibly caused by RTN activation, can only be measured with any reliability during periods when breathing is regular (animals are resting) as opposed to ambulating, sniffing, eating etc...*

An infrared camera was used to record mouse behavior during experiments in the dark/active state. As the reviewer suspects, mice were considerably more active during the dark compared to the light/inactive state so they took a bit longer to acclimate to the chamber or calm down after each gas exposure. However, it was not difficult to find sections of data under each experimental condition where the mouse appeared to be quietly resting. Behavior artifacts were excluded from analysis.

7. Line 212: “and in some cases *Phox2b* immunoreactivity or *Phox2b*-dependent *tdT* expression”. As already mentioned, the presence of *TdT* does not identify RTN neurons specifically; *TdT* could be present in any neuron that has expressed *Phox2b* at any point during development (e.g., facial motoneurons, astrocytes). Also, in neonate mice, *Phox2b* immunoreactivity is still detectable in facial motor neurons. In any event, the authors should report what proportion of presumptive RTN neurons were actually identified using these histological criteria.

Good point. We have added these details to the results section.

8. Line 214-215; many would argue that an increase in firing rate of 1.7 Hz from a baseline of 1.8Hz when CO₂ is increased from 5 to 10%, is anything but “robust” even though it may be typical of RTN neurons in the authors’ experimental conditions. Similar or larger increases have been reported in many other neurons (NTS, serotonin, Locus coeruleus). Does a more alkaline reference point (3% CO₂?) increase the dynamic range of the response to acidification?

To avoid overstatements we omitted ‘robust’. We agree that it would be nice to expand the CO₂/H⁺ response profile to include a point below control CO₂ levels; however, in this work we were not able to make that determination. Nevertheless, it is worth noting that in previous work we showed RTN neurons with a nearly identical CO₂/H⁺ sensitivity as the cells included in this study, were inhibited by 3% CO₂ and showed a CO₂ response profile consistent with type I chemoreceptors described by others (PMID: 19711410).

9. Lines 229-230: “These results show that expression of *Kcnq2R201C* in *Phox2b*+ neurons caused cell autonomous suppression of baseline firing and CO₂/H⁺ stimulated activity”. Why is the effect of this mutation on RTN neurons deemed “cell-autonomous”? This could be true of course but no attempt was made at isolating RTN neuron synaptically. Besides, this and other labs have previously published evidence that, in slices, RTN neurons are at least partly regulated via CO₂-sensitive synaptic inputs (serotonergic, SST-ergic). <https://pubmed.ncbi.nlm.nih.gov/30866045/>; <https://pubmed.ncbi.nlm.nih.gov/34013884/>.

You are correct, we did not block synaptic or paracrine inputs to RTN neurons during these experiments so we have re-worded the statement to read “... expression of *Kcnq2*^{R201C} in *Phox2b*+ neurons suppressed baseline firing and CO₂/H⁺-stimulated activity of chemosensitive RTN neurons”.

10. Lines 481-482: This should likely state that the anesthetized animals were perfused with paraformaldehyde before the brains were removed? or was the RNAscope and IHC performed on unperfused tissue?

Thanks for bringing this to our attention. Animals were perfused for both RNAscope and IHC experiments. We have edited that statement to read “Animals were anesthetized (ketamine, 75 mg/kg; xylazine, 5 mg/kg; IP) then transcardially perfused with 20 mL of room temperature phosphate buffered saline (PBS, pH 7.4) followed by chilled 2% paraformaldehyde (pH 7.4 in 0.1 M PBS) by peristaltic pump. After fixation, brains were removed...”.

Reviewer #2 Comments:

1) Authors responded to all my concerns raised from the previous edition. The experiments that I

suggested turned inconclusive, but it is understandable that it could be failed due to non-selective nature of pharmacological experiments and compound-specific limitation. However, readers deserve to know the finding of ML252 and XE991 in whole animal responses with a discussion of potential causes for negative results. Short statement would be sufficient for this propose.

We thank this reviewer for their support and helpful suggestions.

The ML252 results are a complete data set so we agree that it is reasonable to report these findings. However, the XE991 data (n=2) is too preliminary to have any confidence in these results so we do not feel comfortable commenting on those results at this time.

We added the following to the last paragraph of results “These results also identify Kcnq2 channels as a potential therapeutic target for breathing problems in *Kcnq2* encephalopathy. To test this possibility, we treated control and Kcnq2 GOF mice with a single dose of ML252 (30mg/kg; dissolved in DMSO and diluted in saline for I.P. injection) followed 30 min later by assessment of respiratory function. Unexpectedly, we found this treatment had no discernible effect on breathing in control or KCNQ2 GOF mice (N=6 mice per genotype). However, in the absence of a positive control we interpret these results with caution and remain optimistic that Kcnq2 channels can be targeted by systemic drug application to improve respiratory output”. We also added the following to the limitations section of the discussion “Lastly, although there is some evidence that ML252 delivered I.P. can reach the brain (PMID: 22793372), there is a scarcity of published studies describing effects of ML252 on animal behavior so we are unsure of the drug dosage. Therefore, that finding should be interpreted with caution”.

2a) On the other hand, revised Fig. 1 and Fig. 2 brought new confusions that need to be addressed. Fig. 1. First of all, does the legend explain correctly regarding white and red arrows? The legend indicates “white arrows identify cells that express KCNQ2 but not KCNQ3 and red arrows identify cells that co-express both transcripts” Beside typo (identity), I could not confirm KCNQ2 transcript in RTN cells with white arrows, but only in red arrows. In addition, it is hard to tell KCNQ3 shows any transcripts in all panels. This reviewer thinks authors need to define the criteria for “positive transcript” in the method section. Also, corresponding arrows should be indicated in KCNQ2 and KCNQ3 panels to show where we should look at. Finally, there is a purple arrow in panel B merge. What is it?

Typo fixed.

We have reimaged at a higher magnification in the hopes of making the transcript signals more apparent. Thank you for reminding us to define our criteria for positive transcript; cells with 5 or more puncta in the TdT labeled soma were considered positive for that transcript. This information has been clarified in the methods. We also added arrows in each inset and defined each colored arrow in the legend. To clarify, the purple arrow (now yellow) identifies those few Phox2b+ cells that are positive for KCNQ3 but negative for KCNQ2.

2b) For panel D, total cell counts were reported in Supplementary data set as I requested. However, supplementary dataset has never been mentioned in the main text and should be presented in the relevant section.

We have carefully reviewed the ms and can confirm that all figures and supplemental data are discussed in the relevant sections of text.

2c) In panel E, reported number and what are shown did not match. For instance, for 7N, I can count 14 symbols but reported n was 12.

These numbers have been updated to reflect additional analysis. We confirmed that all slices are accounted for in the fig. Note that due to the dense clustering some dots might be obscured.

3) Fig. 2 panel B. In my initial review, I thought this was genomic PCR, but the legend reads “transcript”, thus suggesting RT-PCR. Is this true? Since Fig. 2A and 2B are barely explained in the main text, it is hard to confirm. This should be clarified. In addition, the revised figure 2B indicates KCNQ2f/f gives shorter band than KCNQ2+/+, which contradicts to the previous edition. Which is correct?

Sorry for the confusion. Your initial impression is correct, this was not RT-PCR. we have corrected the legend as follows “**B**, Genotyping PCR analysis for Kcnq2 GOF and Kcnq2 cKO mice. The PCR products run to the expected sizes for each genotype and primer set (Kcnq2 GOF primers span exon 4, including residue 201 of exon 4; Kcnq2 cKO primers include a loxP site). Water was used as a no template negative control”.

4) Page 8 line 2. After reading comments from reviewer 1, I wonder “diminished RTN chemoreception” should be rephrased to “diminished RTN sensation of CO2” would be more specific for this paper. Please consider.

We appreciate the suggestion. Since we manipulate Kcnq2 function in all Phox2b-expressing cells, we will follow the first reviewers suggestion to limit our interpretation to just that population.

Reviewer #3 Comments:

The authors have satisfactorily addressed the major and minor issues in their revision. Missing methodological details are now provided, statistical methods are adequately described, and figures and figure legends are now clearer. The discussion touches on relevant points, and the authors' conclusions are supported by the data presented.

We thank this reviewer for their support.

REVIEWERS' COMMENTS

Reviewer #1 (Remarks to the Author):

The latest version of this excellent study has addressed all my prior concerns.

Reviewer #2 (Remarks to the Author):

Authors addressed all my concerns sufficiently. I have no more concerns.

Reviewer #1 Comments:

The latest version of this excellent study has addressed all my prior concerns.

We thank the reviewer for their helpful suggestions during the review process.

Reviewer #2 Comments:

Authors addressed all my concerns sufficiently. I have no more concerns.

We thank the reviewer for their helpful suggestions during the review process.